# miR-210 Regulates Apoptotic Cell Death during Cellular Hypoxia and Reoxygenation in a Diametrically Opposite Manner

**DOI:** 10.3390/biomedicines10010042

**Published:** 2021-12-25

**Authors:** Gurdeep Marwarha, Øystein Røsand, Nathan Scrimgeour, Katrine Hordnes Slagsvold, Morten Andre Høydal

**Affiliations:** 1Group of Molecular and Cellular Cardiology, Department of Circulation and Medical Imaging, Faculty of Medicine and Health, Norwegian University of Technology and Science (NTNU), 7030 Trondheim, Norway; gurdeep.marwarha@ntnu.no (G.M.); oystein.rosand@ntnu.no (Ø.R.); nathan.scrimgeour@ntnu.no (N.S.); katrine.h.slagsvold@ntnu.no (K.H.S.); 2Department of Cardiothoracic Surgery, St. Olavs University Hospital, 7030 Trondheim, Norway

**Keywords:** miR-210, hypoxia, AC-16 cardiomyocytes, apoptosis, hypoxia-reoxygenation

## Abstract

Apoptotic cell death of cardiomyocytes is a characteristic hallmark of ischemia–reperfusion (I/R) injury. The master *hypoxamiR*, microRNA-210 (miR-210), is considered the primary driver of the cellular response to hypoxic stress. However, to date, no consensus has emerged with regards to the polarity of the miR-210-elicited cellular response, as miR-210 has been shown to exacerbate as well as attenuate hypoxia-driven apoptotic cell death. Herein, in AC-16 cardiomyocytes subjected to hypoxia-reoxygenation (H-R) stress, we unravel novel facets of miR-210 biology and resolve the biological response mediated by miR-210 into the hypoxia and reoxygenation temporal components. Using transient overexpression and decoy/inhibition vectors to modulate miR-210 expression, we elucidated a *Janus* role miR-210 in the cellular response to H-R stress, wherein miR-210 mitigated the hypoxia-induced apoptotic cell death but exacerbated apoptotic cell death during cellular reoxygenation. We further delineated the underlying cellular mechanisms that confer this diametrically opposite effect of miR-210 on apoptotic cell death. Our exhaustive biochemical assays cogently demonstrate that miR-210 attenuates the hypoxia-driven *intrinsic apoptosis* pathway, while significantly augmenting the reoxygenation-induced caspase-8-mediated *extrinsic apoptosis* pathway. Our study is the first to unveil this *Janus* role of miR-210 and to substantiate the cellular mechanisms that underlie this functional duality.

## 1. Introduction

Ischemic heart disease (IHD) is the leading cause of morbidity and mortality in the world [1,2,3]. The myocardial ischemia-induced chronic hypoxia evokes cardiomyocyte cell death, leading to irreversible damage to the myocardium that manifests clinically as acute myocardial infarction (AMI), a clinical condition characterized by a high rate of mortality when left untreated. Post-AMI prognosis remains bleak due to increased risk for recurrent ischemia leading to the ensuing AMI events, and the subsequent progression into ischemic heart failure (IHF) or ischemic cardiomyopathy. Beneficial prognosis post-AMI is contingent on timely coronary reperfusion of the ischemic myocardium to limit the infarct size. Timely coronary reperfusion is the contemporary standard therapy for AMI, but paradoxically exacerbates tissue injury resulting in an increase in cardiomyocyte death, a pathophysiological phenomenon termed ischemia–reperfusion (I/R) injury, which may culminate in cardiac dysfunction [4,5,6,7,8]. Ergo, designing therapeutic interventions that mitigate the myocardial I/R injury-induced cardiomyocyte death has been the focal point of numerous empirical animal studies and clinical research [7,9]. Evidence from a plethora of contemporary studies widely implicates apoptotic, necrotic, and autophagic cardiomyocyte death in response I/R injury [10]. While the terminal biochemical conduit of cardiomyocyte death has been characterized extensively, the upstream cellular events and the ensuing molecular mechanisms that mediate the propagation of the noxious stimuli and the molecular entities that eventually actuate and effectuate the cardiomyocyte death have not been exhaustively elucidated and delineated. The *hypoxamiR* (hypoxia-induced microRNA) microRNA-210 (miR-210), considered as one of the most significant molecular determinants of the orchestrated compensatory cellular response to myocardial ischemia and cellular hypoxia, has been implicated in the regulation of multiple facets of cellular function that impinge upon pathways that govern cell viability under hypoxic conditions [11,12,13,14,15,16,17]. In the last decade, a substantial volume of laboratory and clinical studies has implicated miR-210 as the most significant modulator of the hypoxia-induced derangements in cell proliferation, cell death, mitochondrial function, DNA repair, and angiogenesis [12,14,16,18,19]. Consequently, miR-210 has been characterized a master *hypoxamiR* [16]. miR-210 has been widely implicated in the regulation of mitochondrial function [20,21,22,23], cell proliferation [24,25,26,27,28,29,30], and the modulation of apoptotic cell death [26,31,32,33,34,35,36,37]—all key facets of cellular function known to be dysregulated upon hypoxia-reoxygenation challenge or I/R injury. Transgenic mice that constitutively overexpress miR-210 exhibit greater compensatory cardiomyocyte cell proliferation, enhanced cardiomyocyte survival, and augmented angiogenesis in established experimental mouse models of I/R injury and AMI [38], lending support to the potential of miR-210 being touted as the novel therapeutic target in IHD [33]. However, subsequent studies have unveiled a more contradictory therapeutic role of miR-210, as given that miR-210 plays an indispensable role in cell proliferation and angiogenesis, it is envisaged that miR-210 could be a mediator of pathological maladaptive cardiac hypertrophic remodeling that leads to IHF [17]. Additionally, there is no consensus as to whether the hypoxia-reoxygenation (H-R) challenge or I/R injury-induced enhanced miR-210 expression has a salutary protective response that augments cell proliferation and cell survival [32,33,36,37] or a detrimental response where it may exacerbate and augment cardiomyocyte cell death [34,39] as well as attenuate cell proliferation [24,40]. Furthermore, there is a significant gap in the understanding of the cellular signaling pathways, the ensuing molecular mediators, and the subsequent temporal kinetics of the H-R-induced miR-210 expression and the consequent impact and bearing this temporal profile may exert on the eventual cardiomyocyte viability. Despite being characterized as a master *hypoxamiR* [16] and overwhelmingly demonstrated to modulate apoptotic cell death in response to cellular hypoxia, a clear consensus has not emerged pertaining to the functional role of miR-210, whether it mediates an adaptive, maladaptive, or perhaps a hybrid response to hypoxia-reoxygenation challenge or I/R injury. This can be attributed to disparities in the experimental paradigm itself, as preponderance of studies have determined the functional role of miR-210 in the hypoxia phase, while very few studies have examined the functional role of miR-210 in the reoxygenation phase. Given that miR-210 programs the cellular machinery to adapt to a hypoxia phenotype [37], it warrants delineating the functional role of miR-210 in the reoxygenation phase that is characterized by a cellular milieu in contradistinction to the hypoxia phase. To address these scientific voids and contemporary deficits that prevail in our understanding of the role of miR-210 in myocardial ischemia-elicited cellular hypoxia, we exhaustively delineated the role of miR-210 mediated cellular response that modulates apoptotic cell death during cellular hypoxia and the ensuing cellular reoxygenation phase. In this study, we dissected and resolved the biochemical impact of miR-210 expression modulation on H-R-induced apoptotic cell death into two distinct temporal components, namely, the hypoxia phase versus the reoxygenation phase, and subsequently delineated the underlying cellular mechanisms.

## 2. Materials and Methods

### 2.1. Cell Culture and Treatments

Human AC-16 cardiomyocyte cells (EMD Millipore/Merck Millipore/Merck Life Sciences, Catalogue # SCC109, Darmstadt, Germany, RRID:CVCL_4U18) were cultured and sub-cultured in the standard *maintenance medium*—Dulbecco’s modified Eagle’s medium (DMEM)/Ham’s F12 (1:1; *v*/*v*) with 2 mM glutamine, 12.5% fetal bovine serum (FBS), and 1% antibiotic/antimycotic mix, in accordance to the standard guidelines, procedures, and protocols established by the commercial vendor. For the *gain-of-function* and the *loss-of-function* experiments, AC-16 cells were *reverse* transfected with the miR-210 expression vector (GeneCopoeia^TM^, Rockville, MD, USA, Catalogue # HmiR0167-MR04) or the miR-210-3p *decoy/inhibitor* vector (GeneCopoeia^TM^, Rockville, MD, USA, Catalogue # HmiR-AN0317-AM01). AC-16 cells were *reverse* transfected with the respective vectors, using Polyfect^®^ (Qiagen Norge, Oslo, Norway, Catalogue # 301107) in accordance to the manufacturer’s guidelines and standardized procedures [41]. The plasmid load to be transfected was standardized to 1 μg per 0.6 × 10^6^ cells and *scaled up* or *scaled down* in accordance to the stipulations of the experimental paradigm [41]. The hypoxia challenge (1% O_2_, 5% CO_2_, and 94% N_2_ for 18 h) was effectuated by incubating the transfected AC-16 cells with the specific *hypoxia medium* (Appendix A) for 18 h, followed by incubation under normoxic conditions (reoxygenation experimental groups) for 8 h with the standard *maintenance medium.* The hypoxia and the hypoxia-reoxygenation (H-R) experimental paradigm is depicted in Table 1. Hypoxia challenge (1% O_2_, 5% CO_2_, and 94% N_2_ for 18 h) was induced and maintained for the designated duration using the *New Brunswick™ Galaxy^®^ 48 R CO_2_ incubator* (Eppendorf Norge AS, Oslo, Norway).

### 2.2. Western Blotting

Mitochondrial and cytosolic fractions from the respective experimental groups were prepared using the mitochondrial fractionation kit “*Cytochrome c Release Apoptosis Assay Kit*” from Sigma Aldrich/Merck Millipore (Merck Life Science, Darmstadt, Germany, Catalogue # QIA87) following the manufacturer’s protocol and guidelines. Proteins (10–50 μg) were resolved on SDS-PAGE (sodium dodecyl sulphate–polyacrylamide gel electrophoresis) gels followed by transfer to a *polyvinylidene difluoride* (PVDF) membrane (Immun-Blot^TM^ PVDF Membrane, Bio-Rad Norway AS, Oslo, Norway, Catalogue # 1620177) [42,43] and overnight incubation with the respective primary antibodies at 4^0^ C following standardized protocols [44,45]. β-Actin was used as a gel loading control for cytosolic fractions, while TOM20 was used as a gel loading control for the mitochondrial fractions. The origin, source, the dilutions of the respective antibodies used in this study is compiled in Table 2. The blots were developed with enhanced chemiluminescence substrate (SuperSignal™ West Pico PLUS Chemiluminescent Substrate, Thermo Fisher Scientific, Oslo, Norway, Catalogue # 34580) and imaged using a LICOR Odyssey Fc imaging system (LI-COR Biotechnology, Cambridge, UK).

### 2.3. Enzyme-Coupled miR-210 Hybridization Immunoassay

The levels of miR-210 in the experimental cell lysates were determined by adopting a novel microRNA immunoassay approach [46,47,48,49,50,51,52,53]. This miR-210 immunoassay approach allowed the direct quantitative determination of miR-210 in the same experimental lysates being subjected to the specific downstream assays. Briefly, miR-210 in the experimental lysates was *affinity-captured* on streptavidin beads by hybridization with biotin-labeled miR-210 *locked nucleic acid (LNA) capture probe* (Qiagen Norge, Oslo, Norway, Catalogue # 339412 YCO0212944). The *affinity-captured* miR-210 was eluted from the streptavidin beads (10 mM Tris, pH 7.5 at 90 °C for 10 min) and un-sequestered from the *double-stranded* hybrid by *denaturation*, and subsequently immobilized in the microwells of a *solid phase 96-well nucleic acid microplate* (Nunc™ NucleoLink™ Strips, Thermo Fisher Scientific, Oslo, Norway, Catalogue # 248259). The immobilized miR-210 was quantitated by adopting an *indirect ELISA* approach [54], whereby the immobilized miR-210 was hybridized with digoxigenin-labeled miR-210 *LNA detection probe* (Qiagen Norge, Oslo, Norway, Catalogue # 339412 YCO0212945) followed by immunodetection with the AP (alkaline phosphatase)-conjugated digoxigenin antibody (*Digoxigenin AP-conjugated Antibody,* R&D Systems, Minneapolis, MN, USA, Catalogue # APM7520) using the AP-substrate PNPP (p-nitrophenyl phosphate, disodium salt) (Thermo Fisher Scientific, Oslo, Norway, Catalogue # 37621) as the chromophore for the colorimetric read-out (λ_405_). Competition assays were also performed with the *unlabeled detection probe* to exhibit *assay specificity* and serve as an *experimental blank*. The raw optical density values measured at λ_05_ (405 nm) were corrected with the *experimental blank* and subsequently normalized and expressed as *fold-change* relative to the experimental control. Data are expressed as a *fold-change* ± standard deviation (S.D) from three technical replicates for each of the four biological replicates belonging to each experimental group (*n* = 4).

### 2.4. Lactate Dehydrogenase (LDH) Assay

The levels of lactate dehydrogenase (LDH) in the conditioned media were determined as a surrogate measure of generic cell death. LDH levels in the conditioned medium were measured by using a *sandwich ELISA immunoassay* approach [55,56,57,58]. Briefly, 20 ng of LDH *capture* antibody (Table 2) was immobilized in each well of a 96-well microplate [59,60,61]. The respective conditioned media (50 μL) from experimental samples were incubated with the immobilized LDH *capture* antibody, overnight at 4 °C. The conditioned media was discarded, the 96-well microplate wells were washed 3× (15 min each) with TBS-T (Tris-buffered saline with 0.1% *v/v* Tween-20) and incubated with the LDH-A and LDH-B *detection* antibodies (Table 2), overnight at 4 °C. The 96-well microplate wells were washed 3× (15 min each) with TBS-T, followed by immunodetection with the HRP (horseradish peroxidase)-conjugated secondary antibody [60] using the HRP-substrate OPD (o-phenylenediamine dihydrochloride) (Thermo Fisher Scientific, Oslo, Norway, Catalogue # 34005) [62,63] as a chromophore for the colorimetric read-out (λ_450_). The antibodies and signal specificity were established by performing *peptide blocking assays* in the entire gamut of experimental lysates. The *LDH-A antibody blocking peptide* (Novus Biologicals/Bio-Techne, Abingdon, United Kingdom, Catalogue # NBP1-48336PEP) and *LDH-B antibody blocking peptide* (Novus Biologicals/Bio-Techne, Abingdon, United Kingdom, Catalogue # NBP2-38131PEP) corresponding to the specific epitopes for the LDH-A and LDH-B antibodies (Table 2), respectively, were used for the *peptide blocking assays*. The respective optical density (O.D) values from the *peptide blocking assays* were used for *experimental blank correction*. Data is expressed as *experimental blank-corrected* O.D_450_ (λ_450_) values from three technical replicates for each of the four biological replicates belonging to each experimental group (*n* = 4).

### 2.5. Caspase-3 Activity Assay

The enzymatic activity of caspase-3 was measured spectrophotometrically using the specific caspase-3 substrate Ac-DEVD-p-NA (n-Acetyl-Ac-Asp-Glu-Val-Asp-p-nitroanilide) (Sigma Aldrich, Oslo, Norway, Catalogue # 235400-5MG). Caspase-3 activity was measured as a surrogate of the abundance of the released chromophore engendered by the proteolysis of Ac-DEVD-p-NA by caspase-3 [64]. Briefly, AC-16 cells—terminally sub-cultured and plated in 100 mm cell-culture plates to the desired confluence (4 × 10^6^ cells per plate) and subjected to the respective transfection and experimental interventions—were trypsinized and pelleted by centrifugation (1000× *g* for 5 min). The pelleted cells were resuspended in the cell lysis buffer (50 mM HEPES, 5 mM CHAPS; pH 7.4) and incubated on ice for 10 min to lyse the cells, followed by centrifugation at 12,000× *g* for 15 min to pellet the cell debris. The supernatant containing the *non-denatured* cell lysates (devoid of protease inhibitors) was dedicated as input for caspase-3 activity determination. The protein content of the *non-denatured* cell lysates was adjusted to 4 µg/µL concentration, and the volume was adjusted to 180 µL with the addition of the assay buffer (20 mM HEPES, 1.62 mM CHAPS, 10 mM Nacl, 2 mM EDTA; pH 7.4). A 200 µg equivalent of protein content (50 µL of the lysate at 4 µg/µL diluted to 180 µL) per well of a 96-well microplate was used as *input* from each experimental sample for the caspase-3 activity assay. The *input* from the respective samples was incubated with the caspase-3 substrate, Ac-DEVD-p-NA (200 µM, 20 µL of 2 mM in a 200 µL assay volume), for 4 h at 37 °C. The caspase-3 activity assays were also performed in the presence of the caspase-3 inhibitor, Ac-DEVD-CHO [64] (Sigma Aldrich/Merck Millipore/Merck Life Science, Darmstadt, Germany, Catalogue # 235420) to exhibit *assay specificity* and serve as an *experimental blank*. The absorbance at λ_405_ (405 nm) corresponding to the amount of the chromophore engendered as a commensurate measure of caspase-3 activity was determined using a microplate reader. The respective absorbances from the caspase-3 inhibitor-treated lysates were used for *experimental blank correction*. Data are expressed as *experimental blank-corrected* O.D_405_ (λ_405_) values from three technical replicates for each of the four biological replicates belonging to each experimental group (*n* = 4).

### 2.6. Terminal Deoxynucleotidyl Transferase dUTP Nick End Labeling (TUNEL) Assay

The magnitude of DNA fragmentation as a morphological hallmark of *late apoptosis* was determined by using the in situ quantitative colorimetric apoptosis detection system “HT TiterTACS™Apoptosis Detection Kit” from R&D Systems (R&D Systems, Minneapolis, MN, USA, Catalogue # 4822-96-K) following the manufacturer’s guidelines and well established contemporary protocols [65,66,67,68]. Briefly, AC-16 cells—terminally sub-cultured and plated in 96-well microplates to the desired confluence (5 × 10^4^ cells/well) and subjected to the respective transfection and experimental interventions (as enunciated earlier)—were fixed in situ, and subsequently *3′-hydroxyl nick-end labeled* with biotin-conjugated dNTPs (deoxynucleotide triphosphates) using the DNA polymerase *terminal deoxynucleotidyl transferase (TdT)* [69]. The *3′-hydroxyl nick-end-*incorporated biotin-conjugated dNTP were detected with streptavidin-conjugated HRP using the HRP-substrate TACS-Sapphire as a chromophore for the colorimetric read-out (λ_450_). The assays performed with *unlabeled* experimental samples, devoid of TdT in the labeling mix, were used to establish signal specificity and serve as *experimental blank correction.* Data are expressed as *experimental blank-corrected* O.D_450_ (λ_450_) values from three technical replicates for each of the four biological replicates belonging to each experimental group (*n* = 4).

### 2.7. Cytochrome c Release Assay

The translocation of *cytochrome c* from the mitochondrial intermembrane space to cytosol was determined by using the mitochondrial fractionation kit “*Cytochrome c Release Apoptosis Assay Kit*” from Sigma Aldrich/Merck Millipore (Merck Life Science, Darmstadt, Germany, Catalogue # QIA87) following the manufacturer’s protocol and guidelines. Briefly, AC-16 cells—terminally sub-cultured and plated in 100 mm cell-culture plates to the desired confluence (4 × 10^6^ cells/plate) and subjected to the respective transfection and experimental interventions (as enunciated earlier)—were trypsinized and pelleted by centrifugation (1000× *g* for 5 min). The pelleted cells were resuspended in the *cytosolic extraction buffer* (supplied with the kit) containing DTT (dithiothreitol) as well as protease inhibitors (Halt™ Protease Inhibitor Cocktail 100x, Thermo Fisher Scientific, Oslo, Norway, Catalogue # 78429) and incubated on ice for 10 min to lyse the cells followed by centrifugation at 12,000× *g* for 30 min, to pellet the cell debris. The resultant supernatant was collected as the *cytosolic fraction*, while the resultant pellet was resuspended in the *mitochondrial extraction buffer* (supplied with the kit), containing DTT as well as protease inhibitors, to generate the *mitochondrial fraction*. The respective fractions were processed for Western blotting analysis following standardized protocols [44,45] to determine the integrity of the respective fractions. The abundance of *cytochrome c* in the respective fractions was measured by using a *sandwich ELISA immunoassay* approach [55,56,57,58]. Briefly, 20 ng of the *cytochrome c capture* antibody (Table 2) was immobilized in each well of a 96-well microplate [59,60,61]. The respective cytosolic fractions (equivalent to 50 μg of protein content) and mitochondrial fractions (equivalent to 20 μg of protein content) were incubated with the immobilized *cytochrome c capture* antibody, overnight at 4 °C. The conditioned cytosolic fractions and mitochondrial fractions were discarded, and the 96-well microplate wells were washed 3× (15 min each) with TBS-T and incubated with the *cytochrome c detection* antibody (Table 2), overnight at 4 °C. The 96-well microplate wells were washed 3× (15 min each) with TBS-T, followed by immunodetection with the HRP-conjugated secondary antibody [60] using the HRP-substrate OPD (Thermo Fisher Scientific, Oslo, Norway, Catalogue # 34005) [62,63] as a chromophore for the colorimetric read-out (λ_450_). The antibody and signal specificity were established by performing *peptide blocking assay* in the entire gamut of experimental lysates. The *cytochrome c antibody blocking peptide* (Cell Signaling Technology, Danvers, MA, USA, Catalogue # 1033) corresponding to the specific epitope for the *cytochrome c* antibody (Table 2) was used for the *peptide blocking assay*. The respective absorbances from the *peptide blocking assay* were used for *experimental blank correction*. Data are expressed as *experimental blank-corrected* O.D_450_ (λ_450_) values from three technical replicates for each of the four biological replicates belonging to each experimental group (*n* = 4).

### 2.8. Co-Immunoprecipitation (Co-IP) Analysis

Co-immunoprecipitation (Co-IP) assays from the respective experimental AC-16 cell homogenates were performed to determine the relative abundance of the *apoptosome complex* and the *DISC-IIa complex.* For the *apoptosome complex* formation assay (Section 2.9), AC-16 cells (1 × 10^6^)—seeded in 100 mm plates, transfected, and subjected to the experimental interventions (as enunciated earlier)—were homogenized using a *non-denaturing* lysis buffer (20 mM Tris, 137 mM Nacl, 2 mM EDTA, 1% Nonidet P-40, 10% glycerol; pH 7.4) supplemented with protease and phosphatase inhibitors. For the *DISC-IIa complex* formation assay (Section 2.12), *RIPK1* (receptor-interacting serine/threonine-protein kinase 1) *precleared lysates* (prepared as described in the Section 2.11) were used as the *input* for the ensuing Co-IP procedure. For the immunoprecipitation procedure, cell lysate containing the equivalent to 750 μg of total protein was pre-cleared by incubation with protein A/G-coated agarose beads for 30 min at 4 °C to reduce the non-specific binding of proteins to the beads. The equivalent of 750 μg of total protein content of the pre-cleared lysate was subsequently incubated with either 5 μg of target antibody or 5 μg of the corresponding control IgG antibody, overnight at 4 °C. The respective immunocomplexes were captured and immobilized by the addition of protein A/G agarose beads and incubation overnight at 4 °C. The beads containing the immunocomplexes were washed 3x with the *non-denaturing* lysis buffer followed by centrifugation and discarder of the supernatant. The beads were suspended in *denaturing* RIPA buffer (50 mM Tris, 150 mM Nacl, 0.1% SDS, 0.5% sodium deoxycholate, 1% Triton X; pH 7.4) supplemented with protease and phosphatase inhibitors and centrifuged to pellet the beads. The supernatant containing the immunoprecipitated proteins was subjected to *tandem* ELISA with the designated antibodies (Table 2).

### 2.9. Apoptosome Complex Formation Assay by Co-Immunoprecipitation (Co-IP) and Tandem ELISA

Co-IP coupled with *tandem* ELISA determining the quantitative abundance of APAF1 (apoptotic peptidase activating factor 1) and *cytochrome c* in the procaspase-9 immunoprecipitates were performed as a surrogate measure of the abundance of the *apoptosome complex* formation. The relative enrichment of APAF1 and *cytochrome c* in the procaspase-9 immunoprecipitates (prepared as described in the aforementioned “*Co-Immunoprecipitation [Co-IP] Analysis*” excerpt in the Materials and Methods section) was determined by using a *Sandwich ELISA immunoassay* approach. Briefly, 30 ng of APAF1 *capture* antibody (Table 2), 20 ng of *cytochrome c capture* antibody (Table 2), and 30 ng of procaspase-9 *capture* antibody (Table 2) were immobilized in the respective wells of a 96-well microplate [59,60,61]. The *inputs* (50 ng) from the procaspase-9 immunoprecipitates were incubated with the respective immobilized *capture* antibodies, overnight at 4 °C. The conditioned cell lysate was discarded, and the 96-well microplate wells were washed 3× (15 min each) with TBS-T and incubated overnight, at 4 °C, with the respective *detection* antibodies, i.e., APAF1 *detection* antibody (Table 2), *cytochrome c detection* antibody (Table 2), and procaspase-9 *detection* antibody (Table 2). The 96-well microplate wells were washed 3× (15 min each) with TBS-T, followed by immunodetection with HRP-conjugated secondary antibody [60] using the HRP-substrate *Amplex Red* (10-acetyl-3,7-dihydroxyphenoxazine) (Thermo Fisher Scientific, Oslo, Norway, Catalogue # A22188) as the chromophore for the colorimetric read-out (λ_570_). The antibody and signal specificity were established by performing *peptide blocking assays* in the entire gamut of experimental lysates. The *APAF1 antibody blocking peptide* (Novus Biologicals/Bio-Techne, Abingdon, United Kingdom, Catalogue # NBP1-77000PEP), *cytochrome c antibody blocking peptide* (Cell Signaling Technology, Danvers, MA, USA, Catalogue # 1033), and the *procaspase-9 antibody blocking peptide* (Novus Biologicals/Bio-Techne, Abingdon, United Kingdom, Catalogue # NBP1-76961PEP), corresponding to the specific epitopes recognized by the respective *detection* antibodies, were used for the *peptide blocking assays*. The respective O.D values from the *peptide blocking assays* were used for *experimental blank correction*. Data are expressed as *experimental blank-corrected* O.D_570_ (λ_570_) values from three technical replicates for each of the four biological replicates belonging to each experimental group (*n* = 4).

### 2.10. Caspase-8 Activity Assay

The enzymatic activity of caspase-3 was measured spectrophotometrically using the specific caspase-8 substrate Ac-IETD-pNA (Acetyl-Ile-Glu-Thr-Asp-p-Nitroaniline) (Sigma Aldrich, Oslo, Norway, Catalogue # A9968-5MG). Caspase-8 activity was measured as a surrogate of the abundance of the released chromophore p-nitroaniline engendered by the proteolysis of Ac-IETD-pNA by caspase-8. Briefly, AC-16 cells—terminally sub-cultured and plated in 100 mm cell-culture plates to the desired confluence (4 × 10^6^ cells per plate) and subjected to the respective transfection and experimental interventions (as enunciated earlier)—were trypsinized and pelleted by centrifugation (1000× *g* for 5 min). The pelleted cells were resuspended in the cell lysis buffer (50 mM HEPES, 5 mM CHAPS; pH 7.4) and incubated on ice for 10 min to lyse the cells, followed by centrifugation at 12,000× *g* for 15 min to pellet the cell debris. The supernatant containing the non-denatured cell lysates (devoid of protease inhibitors) was dedicated as input for caspase-8 activity determination. The protein content of the non-denatured cell lysates was adjusted to 5 µg/µL concentration, and the volume was adjusted to 180 µL with the addition of the assay buffer (20 mM HEPES, 1.62 mM CHAPS, 2 mM EDTA, 5% *w/v* sucrose; pH 7.4). A 300 µg equivalent of protein content (60 µL of the lysate at 5 µg/µL diluted to 180 µL) per well of a 96-well microplate was used as *input* from each experimental sample for the caspase-8 activity assay. The *input* from the respective samples was incubated with the caspase-8 substrate, Ac-IETD-pNA (200 µM, 20 µL of 2 mM in a 200 µL assay volume), for 4 h at 37 °C. The caspase-8 activity assays were also performed in the presence of the caspase-8 inhibitor, Z-IETD-FMK (Z-Ile-Glu[O-ME]-Thr-Asp[O-Me] fluoromethyl ketone) [70] (Sigma Aldrich/Merck Millipore/Merck Life Science, Darmstadt, Germany, Catalogue # 218759) to exhibit *assay specificity* and serve as an *experimental blank*. The absorbance at λ_405_ (405 nm) corresponding to the amount of the chromophore engendered as a commensurate measure of caspase-8 activity was determined using a microplate reader. The respective O.D values from the caspase-8 inhibitor-treated lysates were used for *experimental blank correction*. Data are expressed as *experimental blank-corrected* O.D_405_ (λ_405_) values from three technical replicates for each of the four biological replicates belonging to each experimental group (*n* = 4).

### 2.11. RIPK1 Immunoprecipitation and RIPK1-Precleared Lysate Preparation

RIPK1 was immunoprecipitated out from the experimental native lysates for the *RIPK1-precleared lysate* preparation that served as the *input* for the ensuing Co-IP assays determining *DISC-IIa complex* formation. Briefly, AC-16 cells (1 × 10^6^)—seeded in 100 mm plates, transfected, and subjected to the experimental interventions (as enunciated earlier)—were homogenized using a *non-denaturing* lysis buffer (20 mM Tris, 137 mM Nacl, 2 mM EDTA, 1% Nonidet P-40, 10% glycerol; pH 7.4) supplemented with protease and phosphatase inhibitors. The cell lysate containing the equivalent to 750 μg of total protein content was precleared by incubation with protein A/G coated agarose beads for 30 min at 4 °C to reduce the non-specific binding of proteins to the beads. The equivalent of 750 μg of the precleared lysate was incubated separately, with either 10 μg of the RIPK1 antibody (Table 2) or 10 μg of the control rabbit IgG antibody (Table 2), overnight at 4 °C. The respective immunocomplexes were captured and immobilized by the addition of protein A/G agarose beads and incubation overnight at 4 °C. The beads containing the immunocomplexes were washed 3× with the *non-denaturing* lysis buffer, followed by centrifugation of RIPK1 containing protein A/G beads. The RIPK1 containing protein A/G bead-pelleted fraction was discarded, and the supernatant fraction was collected, which represented the *RIPK1 precleared lysate*. The validity of the *RIPK1 precleared lysates* was confirmed by *sandwich ELISA immunoassay* as follows. Briefly, 20 ng of the RIPK1 *capture* antibody (Table 2) was immobilized in each well of a 96-well microplate [59,60,61]. The respective experimental cell lysates (equivalent to 50 μg of protein content) were incubated with the immobilized RIPK1 *capture* antibody, overnight at 4 °C. The conditioned cell lysate was discarded, and the 96-well microplate wells were washed 3× (15 min each) with TBS-T and incubated with the RIPK1 *detection* antibody (Table 2), overnight at 4 °C. The 96-well microplate wells were washed 3× (15 min each) with TBS-T, followed by immunodetection with the HRP-conjugated secondary antibody [60] using the HRP-substrate TMB (3,3′,5,5′-tetramethylbenzidine) (Thermo Fisher Scientific, Oslo, Norway, Catalogue # N301) as a chromophore for the colorimetric read-out (λ_450_). The antibody and signal specificity were established by performing *peptide blocking assay* in the entire gamut of experimental lysates. The *RIPK1 antibody blocking peptide* (Sigma Aldrich/Merck Millipore/Merck Life Science, Oslo, Norway, Catalogue SBP3500420) corresponding to the specific epitope recognized by the RIPK1 *detection* antibody (Table 2) was used for the *peptide blocking assay*. The respective O.D values from the *peptide blocking assays* were used for *experimental blank correction*. Data are expressed as *experimental blank-corrected* O.D_450_ (λ_450_) values from three technical replicates for each of the four biological replicates belonging to each experimental group (*n* = 4).

### 2.12. DISC-IIa Complex Formation Assay by Co-Immunoprecipitation (Co-IP) and Tandem ELISA

Co-IP assays determining the abundance of procaspase-8 in FADD immunoprecipitates from *RIPK1-precleared lysates* as well as c-FLIP enrichment in procaspase-8 immunoprecipitates from *RIPK1-precleared lysates* were performed as a surrogate measure of the abundance of the *DISC-IIa complex* formation. The relative enrichment of procaspase-8 and c-FLIP in FADD immunoprecipitates from *RIPK1-precleared lysates* (prepared as described in the aforementioned “*Co-Immunoprecipitation (Co-IP) Analysis*” excerpt in the Section 2.8) was determined by using a *sandwich ELISA immunoassay* approach. Briefly, 40 ng of procaspase-8 *capture* antibody (Table 2), 20 ng of c-FLIP *capture* antibody (Table 2), or 20 ng of FADD *capture* antibody (Table 2) was immobilized in the respective wells of a 96-well microplate [59,60,61]. The *inputs* (50 ng), i.e., FADD immunoprecipitates from *RIPK1-precleared lysates*, were incubated with the respective immobilized antibodies, overnight at 4 °C. The conditioned cell lysate was discarded, and the 96-well microplate wells were washed 3× (15 min each) with TBS-T and incubated overnight, at 4 °C, with the respective *detection* antibodies, i.e., procaspase-8 *detection* antibody (Table 2), c-FLIP *detection* antibody (Table 2), and FADD *detection* antibody (Table 2). The 96-well microplate wells were washed 3× (15 min each) with TBS-T, followed by immunodetection with the HRP-conjugated secondary antibody [60] using the HRP-substrate TMB (Thermo Fisher Scientific, Oslo, Norway, Catalogue # N301) as a chromophore for the colorimetric read-out (λ_450_). The antibody and signal specificity were established by performing *peptide blocking assays* in the entire gamut of experimental lysates. The *procaspase-8 antibody blocking peptide* (Novus Biologicals/Bio-Techne, Abingdon, United Kingdom, Catalogue # NBP1-76610PEP), *c-FLIP antibody blocking peptide* (Novus Biologicals/Bio-Techne, Abingdon, United Kingdom, Catalogue # NBP1-77016PEP), and *FADD antibody blocking peptide* (VWR International AS, Oslo, Norway, Catalogue # 10005-492), corresponding to the specific epitopes recognized by the respective *detection* antibodies, were used for the *peptide blocking assays*. The respective absorbances from the *peptide blocking assays* were used for *experimental blank correction*. Data are expressed as *experimental blank-corrected* O.D_450_ (λ_450_) values from three technical replicates for each of the four biological replicates belonging to each experimental group (*n* = 4). Please find simplified outline of the methodological workflow in Figure 1.

### 2.13. Statistical Analysis

The significance of differences among the samples was determined by *one-way analysis of variance* (one-way ANOVA) followed by Tukey’s post hoc test. Statistical analysis was performed with GraphPad Prism 8 (GraphPad Software, San Diego, CA, USA). Quantitative data for all the assays are presented as mean values ± S.D (mean values ± standard deviation).

## 3. Results

### 3.1. miR-210 Mitigated Apoptotic Cell Death during Hypoxia and Enhanced Apoptotic Cell Death during the Reoxygenation Phase

We first determined the potential role of miR-210 in the modulation of generic cell death in response to hypoxia and subsequent reoxygenation (H-R). To this end, we determined LDH release in the conditioned medium emanating from AC-16 cells transfected with the miR-210 expression vector and miR-210-3p decoy/inhibitor vector, at the end of the hypoxia phase (H = 18 h) as well as at the end of the subsequent reoxygenation phase (H-R = 18 h + 8 h). The measurement of LDH levels is considered an accurate surrogate measure of generic cell death [71,72]. The hypoxia-induced increase in generic cell death, as determined by LDH release into the conditioned medium, was significantly attenuated in AC-16 cells overexpressing the endogenous miR-210 (Figure 2A), while significantly being enhanced in AC-16 cells expressing the miR-210-3p decoy/inhibitor (Figure 2B). However, when cell death was assessed at the end of the hypoxia-reoxygenation (H-R) phase (18 h + 8 h), diametrically opposite effects were observed. The H-R-induced increase in generic cell death, as determined by LDH release into the conditioned medium, was significantly enhanced in AC-16 cells overexpressing the endogenous miR-210 (Figure 2C), while significantly being mitigated in AC-16 cells ectopically expressing the miR-210-3p decoy/inhibitor (Figure 2D). Taken together, these data suggest the bimodal nature of miR-210 in the regulation of cell death response that is characterized by conferring refractoriness to hypoxia-induced cell death but paradoxically exacerbating the H-R-induced cell death. We next determined whether the specific modulation of apoptosis-induced cell death was involved in the miR-210-regulated modulation of cell death response during hypoxia and the subsequent H-R insult. To this end, we determined specific molecular and functional markers that characterize apoptotic cell death. Caspase-3 activation is the key regulatory event that instigates the terminal effector events in the apoptotic cascade [64,73,74,75]. We measured caspase-3 activity in non-denatured native lysates emanating from AC-16 cells either overexpressing the endogenous miR-210 or expressing the miR-210-3p decoy/inhibitor, at the end of hypoxia phase (H = 18 h), as well as at the end of the subsequent reoxygenation (H-R = 18 h + 8 h) phase. Overexpression of endogenous miR-210 significantly reduced the hypoxia-induced increase in caspase-3 activity (Figure 3A), while it significantly enhanced the caspase-3 activity during the subsequent H-R insult (Figure 3C). In the corollary experiment, we observed the reciprocal phenomenon, where quenching miR-210-3p activity using the miR-210-3p decoy/inhibitor precipitated a significant enhancement in the hypoxia-induced increase in *caspase-3* activity (Figure 3B), while evoking a significant attenuation in the subsequent H-R challenge-induced increase in *caspase-3* activity (Figure 3D). DNA fragmentation is a cardinal morphological hallmark of late-stage apoptosis. We determined DNA fragmentation by the contemporary standard terminal deoxynucleotidyl transferase dUTP nick end labeling (TUNEL) method [69] adapted to an in situ quantitative colorimetric detection in a 96-well microplate [65,66,67,68]. Overexpression of endogenous miR-210 significantly reduced the hypoxia-induced DNA fragmentation (Figure 4A), while it significantly enhanced the DNA fragmentation during the subsequent H-R insult (Figure 4C). In the corollary experiment, ectopic expression of the miR-210-3p decoy/inhibitor elicited a significant augmentation in the hypoxia-induced DNA fragmentation (Figure 4B), while evoking a significant reduction in the subsequent H-R challenge-induced DNA fragmentation (Figure 4D). Taken together, these data suggest the *Janus* nature of miR-210 in the modulation of apoptotic cell death, whereby miR-210 mitigates the hypoxia-induced apoptotic cell death but paradoxically exacerbates the H-R-induced apoptotic cell death.

### 3.2. miR-210 Regulated the Intrinsic Apoptosis Pathway in a Diametrically Opposite Manner during the Hypoxia Phase and Reoxygenation Phase

We characterized the basis for miR-210-mediated, diametrically opposite modulation of *caspase-3* activity and the ensuing apoptotic cell death during the hypoxia and reoxygenation phases. We focused on the *intrinsic apoptosis pathway* characterized by the release of the mitochondrial protein *cytochrome c* into the cytosol, a cellular event that is indispensable in the formation of the apoptosome complex, which subsequently effectuates *caspase-3* activation. We determined the extent of *cytochrome c* release from the mitochondrial intermembrane space into the cytosol and the subsequent integration of *cytochrome c* with the cytosolic proteins *APAF1* and *procaspase-9*, a characteristic hallmark of the apoptosome ternary complex [76]. To this end, we fractionated and segregated the cytosolic and mitochondrial compartments, and subsequently determined cytochrome c abundance in the respective fractions by ELISA. The integrity and validity of the respective fractions, i.e., cytosolic and mitochondrial, was determined by Western blotting (Appendix A). Overexpression of the endogenous miR-210 induced a significant mitigation in the hypoxia-induced *cytochrome c* release into the cytosol at the hypoxia endpoint (Figure 5A), while eliciting a significant augmentation in the H-R-induced increase in *cytochrome c* release into the cytosol (Figure 5B). The corollary experiment with the ectopic expression of the miR-210-3p decoy/inhibitor concurred with this biphasic role of miR-210 in the regulation of *cytochrome c* release into the cytosol during the hypoxia and reoxygenation phases. miR-210-3p inhibitor evoked a significant enhancement in the hypoxia-induced cytochrome *c* release into the cytosol (Figure 5C), while eliciting a significant attenuation in the H-R-induced increase in *cytochrome c* release into the cytosol (Figure 5D). The release of *cytochrome c* in the cytosol is the rate-limiting step in the apoptosome formation that culminates in the activation of the caspase cascade [76,77]. We therefore determined the extent of cytochrome c “seeding” or “integration” into the apoptosome ternary complex containing the two other requisite components, *APAF1* and *procaspase*-9. To this end, we performed rigorous co-immunoprecipitation (Co-IP) assays as a surrogate measure of the apoptosome formation. We immunoprecipitated *procaspase-9* from the non-denatured lysates and determined the abundance of APAF1 and *cytochrome c* in the *procaspase-9* immunoprecipitates. Consistent with the findings of the role of miR-210 in the regulation of *cytochrome c* release, we found that the overexpression of the endogenous miR-210 significantly attenuated the hypoxia-induced increase in the incorporation of the *APAF1* (Figure 6A inset c) and *cytochrome c* (Figure 6B inset c), along with *procaspase-9* into the ternary apoptosome complex, while ectopic expression of the miR-210-3p decoy/inhibitor significantly exacerbated the hypoxia-induced increase in the incorporation of *APAF1* (Figure 6C inset c) and *cytochrome c* (Figure 6D inset c), along with procaspase-9 into the ternary apoptosome complex. Equitable fractions of *procaspase-9* were immunoprecipitated in all the experimental groups (Figure 6E,F inset c). However, the reciprocal phenomenon was observed with H-R challenge, whereby the overexpression of endogenous miR-210 significantly augmented the H-R-induced increase in the incorporation of APAF1 (Figure 7A inset c) and *cytochrome c* (Figure 7B inset c) along with *procaspase-9* into the ternary apoptosome complex, while the ectopic expression of the miR-210-3p decoy/inhibitor significantly mitigated the H-R-induced increase in the incorporation of APAF1 (Figure 7C inset c) and *cytochrome c* (Figure 7D inset c) along with *procaspase-9* into the ternary *apoptosome* complex. Equitable fractions of *procaspase-9* were immunoprecipitated in all the experimental groups (Figure 7E,F inset c). Respective rabbit-IgG immunoprecipitates were analyzed as the requisite controls to demonstrate the quality and specificity of the *procaspase-9* immunoprecipitates (Figure 6A–F inset b and Figure 7A–F inset b). The validity of miR-210 overexpression and miR-210 knockdown in the respective input lysates was determined by miR-210 hybridization immunoassay (Appendix A).

### 3.3. miR-210 Regulated the Extrinsic Apoptosis Pathway during the H-R Phase but Not during the Hypoxia Phase

We next determined whether miR-210 regulates the *extrinsic apoptosis pathway*. To this end, we first determined *caspase-8* activity as a surrogate marker of *extrinsic apoptotic pathway* activation [78,79] under hypoxia and H-R challenge in the context of miR-210 expression modulation. Hypoxia did not elicit any changes in caspase-8 activity (Figure 8A,B), while H-R challenge evoked a significant increase in caspase-8 activity (Figure 8C,D). At the hypoxia endpoint, neither the overexpression of endogenous miR-210 nor the ectopic expression of the miR-210-3p decoy/inhibitor induced any changes in caspase-8 activity (Figure 8A,B). However, overexpressing endogenous miR-210 induced a significant increase in *caspase-8* activity following H-R (Figure 8C), while the ectopic expression of the miR-210-3p decoy/inhibitor significantly decreased caspase-8 activity following H-R (Figure 8D). We subsequently determined the abundance of the complex IIa of the death-inducing signaling complex *(DISC)* formation, a molecular hallmark of the *extrinsic apoptosis pathway* [80,81], under hypoxia and H-R challenge in the context of miR-210 expression modulation. *DISC* is a plasma membrane-bound heterogenic entity comprising of “death receptor” family of apoptosis-inducing cellular receptors (for example, TNFR, FasR/CD95/DR5) that transduce extracellular apoptotic signals (for example, TNFα, FasL) to intracellular effectors, such as *caspase-8* [78,82], culminating in the activation of the caspase cascade [70,80,83]. The plasma membrane-bound *DISC* can mature and evolve into different internalized cytosolic signaling complexes (complex IIa, complex IIb (ripoptosome), and complex IIc (necrosome)) on the basis of the unique composition of the constituent proteins, as well as their stoichiometric ratios [84,85]. *Procaspase-8* is seeded into all the three aforementioned internalized cytosolic *DISC* signaling platforms [78,79,86]. Therefore, the seeding of *caspase-8* into the *DISC* does not exclusively govern the *extrinsic apoptosis pathway* and could be involved in other forms of programmed cell death response [79,87,88]. It is now consensus that the molecular composition as well as the stoichiometry of the components of *DISC* dictate the ensuing signaling pathway and consequently the mode of cell death, whether through *extrinsic apoptosis* pathway or necroptotic pathway [89,90,91]. By extrapolation of the contemporary molecular understanding of the *caspase-8*-mediated cell death, *caspase-8* activity in *DISC IIa* could be considered as a bona fide marker of the flux through the *extrinsic apoptotic* pathway. The absence of *RIPK1* is the major distinguishing molecular signature of *DISC IIa* relative to *DISC IIb* and *DISC IIc*. As the molecular composition of the *DISC IIa* is homologous to *DISC IIb* and *DISC IIc*, with the signatory absence of *RIPK1* in the *DISC IIa*, we adopted a native sequential co-immunoprecipitation (Co-IP) approach to first deplete the native lysates of *RIPK1* by immunoprecipitation followed by subjecting the *RIPK1-*precleared lysate to tandem immunoprecipitation with the components of *DISC IIa*, namely, *procaspase-8, FADD,* and *c-FLIP*. To this end, we first immunoprecipitated RIPK1 from the native lysates and subsequently subjected the *RIPK1*-depleted native lysates (hereafter designated as *RIPK1-*precleared lysates) as the input for the terminal immunoprecipitation with the adaptor protein *FADD*. The integrity of the *RIPK1-precleared lysates* was corroborated by ELISA (Appendix A). We subsequently immunoprecipitated FADD from the *RIPK1-precleared lysates* and determined the abundance of *procaspase-8* and *c-FLIP*. The overexpression of endogenous miR-210 significantly increased (Figure 9A inset c) while the ectopic expression of the miR-210-3p decoy/inhibitor significantly decreased (Figure 9B inset c) the recruitment of *procaspase-8* into the *FADD* adaptor protein complex during the H-R challenge. The catalytically inactive homolog of *caspase-8* and the master anti-apoptotic protein, *cellular FLICE (FADD-like IL-1β-converting enzyme)-inhibitory protein (c-FLIP)*, is the most significant negative regulator of *caspase-8* activity and the *DISC*-activated *extrinsic apoptosis* pathway [92,93,94,95,96,97,98,99,100,101]. We therefore determined the seeding of *c-FLIP* with *procaspase-8* into the *DISC-IIa complex* [102,103] in *RIPK1-*precleared native lysates. Neither the overexpression of endogenous miR-210 (Figure 9C inset c) nor the ectopic expression of the miR-210-3p decoy/inhibitor (Figure 9D inset c) elicited any effect on the H-R-induced decrease in the recruitment of *c-FLIP* into the *FADD* adaptor protein complex. These data suggest that miR-210 increases *procaspase-8* seeding into the *DISC-IIa complex*, as well as increases the *caspase-8* activity, by virtue of increasing the *procaspase-8*/*c-FLIP* stoichiometric ratio in the *DISC-IIa complex.* The abundance of *FADD* immunoprecipitated in all the experimental groups was determined as the normalizing control (Figure 9E,F inset c). Respective mouse-IgG immunoprecipitates were analyzed as the requisite controls to demonstrate the quality and specificity of the *FADD* immunoprecipitates (Figure 9A–F inset b). The validity of miR-210 overexpression and miR-210 knockdown in the respective *input* lysates was determined by miR-210 hybridization immunoassay (Appendix A). In an integrated overview, our data suggest that miR-210 attenuates the hypoxia-driven apoptotic cell death by inhibiting the hypoxia-induced *intrinsic apoptosis* pathway while augmenting the H-R-induced apoptotic cell death by exacerbating the H-R-induced, both *extrinsic apoptosis* and *intrinsic apoptosis* pathways. The overall inferences drawn from our data, representing the findings and observations of the biphasic role of miR-210 overexpression, are summated and presented as an illustrative model in Figure 10.

## 4. Discussion

Cellular response to I/R injury is inherently a bimodal phenomenon, characterized by transient adaptive responses during ischemia that subsequently reprogram the intracellular molecular milieu to an exorbitant greater susceptibility to reperfusion damage upon clinical and therapeutic intervention. Therefore, it behooves to resolve and dissect the total process leading to cellular injury in response to I/R injury into its respective components, namely, ischemic (hypoxia) component and the reperfusion (reoxygenation) component [104]. Ergo, our study delved into miR-210 modulation of apoptotic cell death, both during hypoxic challenge as well as reoxygenation challenge. Our experimental approach to resolve the miR-210 modulation of H-R-induced apoptotic cell death was spurred by the prevailing gaps in contemporary knowledge. Despite being characterized as a *master hypoxamiR* for more than a decade, a clear consensus has not emerged pertaining to the polarity of the functional effect of miR-210 in the regulation of H-R-induced apoptotic cell death. This lack of consensus pertaining to the functional effect of miR-210 in the regulation of H-R-induced apoptotic cell death is further compounded by preponderance of observations emanating from oncological studies that have ascribed a pleiotropic, but diametrically heterogenous, functional profile of augmented miR-210 expression in the tumor-orchestrated hypoxia microenvironment [29,35]. Our current study partially resolved this functional dichotomy of miR-210 that prevails in contemporary literature by unveiling a *Janus* role of miR-210 in the regulation of apoptotic cell death, one that is defined by the *prevailing temporal phase,* that is, hypoxia versus reoxygenation. Our findings support a bimodal effect of miR-210 on apoptotic cell death, with miR-210 attenuating apoptotic cell death during hypoxia and exacerbating apoptotic cell death during the reoxygenation phase. Adding further complexity to *the bimodal* modulation of apoptotic cell death during hypoxia versus reoxygenation is our unprecedented finding characterizing the underlying biochemical mechanism. Our findings strongly implicate miR-210-modulated inhibition of the *intrinsic apoptosis* pathway during hypoxia and miR-210-modulated augmentation of the *extrinsic apoptosis* pathway during reoxygenation as the biochemical conduit for the respective effects. In the broader context of cell death as a consequence of cellular hypoxia and cellular reoxygenation, our study is narrower and perhaps unidimensional, as it determined only apoptotic cell death in the context of miR-210 expression modulation. Apoptotic cell death is considered a significant contributor to the integrated cellular death observed during cellular hypoxia [105,106,107] and myocardial ischemia [108,109,110,111,112], as well as during cellular reoxygenation and myocardial reperfusion injury [111,112,113,114,115,116,117]. Cardiomyocyte death following chronic myocardial ischemia and myocardial reperfusion injury is attributed to the activation of both the *intrinsic apoptosis* pathway [118,119] and the *extrinsic apoptosis* pathway [120,121]. The findings and observations from our study have conferred miR-210 with a unique disposition, whereby miR-210 regulates both the *intrinsic apoptosis* pathway as well as the *extrinsic apoptosis* pathway, but in a diametrically opposite polarity during the hypoxia and reoxygenation phases. These are unprecedented findings with a profound potential for clinical implications, as enhanced miR-210 expression prevents ischemic tissue damage [122] as well as promoting cardiomyocyte survival and cardiac repair after MI [38]. A plethora of laboratory studies have cogently demonstrated the salutary role of augmented miR-210 expression in increasing cardiomyocyte cell survival, attenuating cardiomyocyte cell death and fostering angiogenesis and cardiac repair, thereby touting miR-210 as a viable therapeutic target [33]. However, findings from our study ascribe a relatively qualified beneficial profile to miR-210, one that is defined by the phase-dependent (hypoxia versus reoxygenation) *temporal* characteristics and kinetics of miR-210 expression. Our unprecedented observations suggest that the hypoxia-induced miR-210 is a bona fide salutary adaptive mechanism that confers the cells with a certain degree of refractoriness towards hypoxia-induced apoptotic cell death. However, under reoxygenation conditions, elevated miR-210 expression enhances the susceptibility of the cells to reoxygenation-induced apoptotic cell death. This duality in the miR-210-regulated apoptotic cell death could be ascribed to the observations that miR-210 inhibits the *intrinsic apoptosis pathway* during hypoxia while fostering the *extrinsic apoptosis pathway* activation upon cellular reoxygenation. This novel and previously unidentified phase-specific (hypoxia vs. reoxygenation) functional duality exhibited by miR-210 could be the underlying attributing phenomenon that may have precipitated the *discord* in the prevailing contemporary literature, which consequently harbors the diametrically opposite functional profile of miR-210, i.e., whether miR-210 has a salutary protective response that augments cell proliferation and cell survival [32,33,36,37], or a detrimental response, where it may exacerbate and augment cardiomyocyte cell death [34,39].

Our study is the first to dissect the effects of miR-210, on the modulation of apoptotic cell death, into the respective phase components (hypoxia vs. reoxygenation), and therefore represents a wider spectrum of the repertoire of miR-210 biological effects that eventually integrates, harmonizes, and resolves the existing dichotomy pertaining to the miR-210-modulated apoptotic cell death response. From the findings and observations reported in this study, we speculate by extrapolation that the *intrinsic apoptosis pathway* predominates during the hypoxia challenge and that cellular reoxygenation reprograms the cellular milieu to a relatively more permissive *extrinsic apoptosis pathway* activation status, thereby precipitating the *diametrically* opposite effect of miR-210 during cellular reoxygenation relative to the effect of miR-210 during hypoxia. Further studies are warranted to delve deeper into the molecular mechanisms that could mediate the dichotomous polar effect of miR-210 on apoptotic cell death during cellular reoxygenation relative to the effect during hypoxia. The serine/threonine kinase Akt is considered one of the most significant modulators of apoptotic cell death and cell proliferation [123,124,125,126]. Emerging evidence has implicated miR-210 in evoking Akt activation and thereby inhibiting apoptosis, fostering cellular proliferation and metastasis, albeit in clinical tumor specimens and primary cancer cells from patient-derived xenografts [127,128]. Cellular inflammation is an important pathophysiological component of the paradoxical cellular injury and cell death observed during H-R and I/R injury [129,130,131,132]. The cellular H-R and pathological I/R injury-induced increase in pro-inflammatory cytokine expression invokes *DISC* (also termed death receptor pathway) activation [133,134,135,136,137], leading to the initiation of the *extrinsic apoptosis* cascade in the cardiomyocytes [129,130,131,132]. Emerging evidence has implicated enhanced miR-210 expression in inducing a proinflammatory phenotype in different cell types [138,139]. Further studies are warranted to address this specific gap in knowledge by delving into the effects of H-R-induced miR-210 expression on proinflammatory cytokine expression and the ensuing activation of the death receptor pathway.

In the context of the findings reported by our study, it is important to appreciate that previous studies have implicated miR-210 in the regulation of apoptotic cell death and cell proliferation, albeit none of the studies discovered a temporal phase-dependent biphasic Janus role of miR-210 in the regulation of apoptotic cell death. Furthermore, preponderance of these studies has examined the role of miR-210 in the regulation of apoptotic cell death in the context of other pathophysiological paradigms and determined specific direct targets of miR-210 in the respective context. In rat myocardial H9C2 cells, miR-210 has been shown to significantly attenuate the hypoxia-induced apoptotic cell death by silencing the *CASP8AP2* (caspase 8-associated protein 2) transcript levels through directly targeting the *CASP8AP2* 3′UTR [140]. However, another study utilizing H9C2 cells subjected to in vitro sepsis experimental challenge concluded that miR-210 overexpression augmented apoptotic cell death by directly binding to the 3′ UTR of the *NDUFA4* (NADH-ubiquinone oxidoreductase MLRQ subunit) mRNA transcript, leading to mitochondrial dysfunction [141]. Moreover, in Neuro-2a neuroblastoma cells, miR-210 exacerbates the hypoxia-induced apoptotic cell death through the directly targeting the Bcl-2 3′UTR [34]. miR-210 also induces apoptotic cell death in human aortic endothelial cells (HAEC) by inhibiting the PI3K-Akt-mTOR signaling pathway through the direct repression of PDPK1 (3-phosphoinositide-dependent protein kinase 1) expression levels by directly targeting the *PDPK1* 3′UTR [142]. In a stark contrast, in U87MG and U251 glioblastoma multiforme (GBM) cell lines, miR-210 induces cell proliferation and inhibits apoptotic cell death by directly decreasing the translation of *ROD1* (regulator of differentiation 1) and *SIN3A* (SIN3 transcription regulator family member A) mRNA transcripts [27,143]. In mouse spermatocyte GC2 cells subjected to hypoxia, miR-210 exacerbates hypoxia-induced apoptotic cell death by directly repressing *KLF7* (Krüppel-like factor 7) expression levels, thereby mitigating the KLF7 transcriptional response, known to confer resistance to hypoxia-driven apoptotic cell death [144]. However, miR-210 expression increases resistance to hypoxia-induced cell death in PC12 cells through the silencing of *BNIP3* (Bcl-2 adenovirus E1B 19 kDa-interacting protein) expression via direct targeting of the *BNIP3* 3′UTR [145]. Furthermore, miR-210 inhibits hypoxia-induced apoptosis of VSMC (vascular smooth muscle cells) via direct targeting of the myogenic transcription factor, *MEF2C* (myocyte enhancer factor 2C) [146]. In HeLa cells, hypoxia-induced inhibition of cell proliferation is ascribed to miR-210-mediated downregulation of a multitude of mitosis-related gene transcripts, including *Plk1* (polo-like kinase 1), *Cdc25b* (cell division cycle 25B), *Cyclin F*, *Bub1b* (BUB1 mitotic checkpoint serine/threonine kinase B), and *Fam83d* (family with sequence similarity 83 member D), by directly targeting their respective 3′-UTR [147]. Additionally, miR-210 expression decreases the proliferation of mouse AB 2.2 embryonic stem cells (AB 2.2 ESC) by silencing Shh (sonic hedgehog) expression via direct targeting of the *Shh* 3′UTR [148]. However, in stark contrast, miR-210 increases cell proliferation in human osteosarcoma cell lines by modulating the Akt-mTOR signaling pathway by directly silencing PIK3R5 (phosphoinositide-3-kinase regulatory subunit 5) expression through directly targeting the *PIK3R5* 3′UTR [127].

### 4.1. Limitations

The findings reported in this study exclusively emanate from immortalized AC-16 cardiomyocyte cells. While AC-16 cardiomyocyte cells are widely considered an excellent in vitro model system to study gene expression changes and the molecular mechanisms that are involved in the regulation of cell survival and cell death pathways [149], they do not fully recapitulate and exhibit the *myogenic phenotype* of the in situ cardiomyocytes and other human primary cardiomyocytes. Furthermore, analogous to other immortalized cell lines, AC-16 cardiomyocyte cells may develop inherent fluxes and divergence from their native functions and cellular properties, and therefore could exhibit an altered adaptive response to both physiological as well as noxious stimuli such as H-R challenge. When we consider our unprecedented findings reported in this study, questions still abound pertaining to the effects of miR-210 modulation on cell proliferation [150,151], especially during the reoxygenation phase. Much of the contemporary focus and interest has been garnered around recent observations of compensatory innate cardiac repair and cardiac regeneration in several mouse models of AMI [152,153]. This postnatal cardiac repair and cardiac regeneration has been ascribed to the resident cardiac stem-cell proliferation and differentiation into adult cardiomyocytes [154], as well as steady-state basal turnover of adult cardiomyocytes [155,156,157]. Further studies are therefore warranted to delve into the impact of miR-210 expression modulation on cardiomyocyte proliferation in cellular models of H-R and animal models of AMI. This study was designed with a broad objective to exhaustively characterize the role of miR-210 in modulating apoptotic cell death and further dissect the phase-dependent response. Our study is unidimensional in exclusively determining and characterizing the role of miR-210 in modulating apoptotic cell death. We took a streamlined approach characterizing apoptotic cell death only, in response to miR-210 expression modulation. Ergo, the impact of miR-210 expression modulation on the hypoxia and other H-R-induced forms of programmed cell death must be determined, and the underlying cellular and molecular mechanisms unveiled. Further studies are therefore warranted to characterize the role of miR-210 in modulating *necroptotic* and the inflammasome-mediated *pyroptotic* cell death in response to hypoxia and H-R challenge.

### 4.2. Future Directions

Although a multitude of miR-210 target genes have been identified and exhaustively validated in the regulation of cell proliferation and apoptotic cell death, our understanding and appreciation of the entirety of the spectrum of the biological effects of miR-210 is plagued by the lack of an integrated approach to characterize the miR-210 *regulome*. The miR-210 *regulome* constitutes the modulated distal indirect targets that are regulated by the miR-210 *targetome* and serve as the effector molecules in eliciting the given cellular, molecular, and biochemical response. This lack of our contemporary knowledge pertaining to the miR-210 *regulome* is the key challenge and major obstacle to overcome in near future. Future studies need to be conceived, designed, and formulated to address this prevailing chasm in our understanding of the miR-210 *regulome*. One approach could be designing studies that utilize a Network Systems multiomics platform-based approach to identify and map the entire miR-210 *regulome*. Data from the miR-210 *regulome* mapping studies would further unveil a targeted strategy to adopt a direct individual transcriptomics, proteomics, and metabolomics-based approaches to elucidate and delineate the underlying cellular and molecular mechanisms.

## 5. Conclusions

Our current study unveiled a unique unprecedented facet of miR-210 with regards to apoptotic cell death under hypoxia and hypoxia-reoxygenation stress. The findings and observations reported in this study provide an added insight into our contemporary comprehension of the vast repertoire of cellular responses elicited by miR-210 in response to hypoxia and hypoxia-reoxygenation stress. Furthermore, the findings from this study have opened unchartered avenues and unveiled potential molecular entities that are the focal point of our ongoing research work and future studies. The findings from this study posit miR-210 as the focal-point of the molecular mechanisms that could be targeted in our quest to design therapeutic interventions to mitigate the ravages of IHD.

## Figures and Tables

**Figure 1 biomedicines-10-00042-f001:**
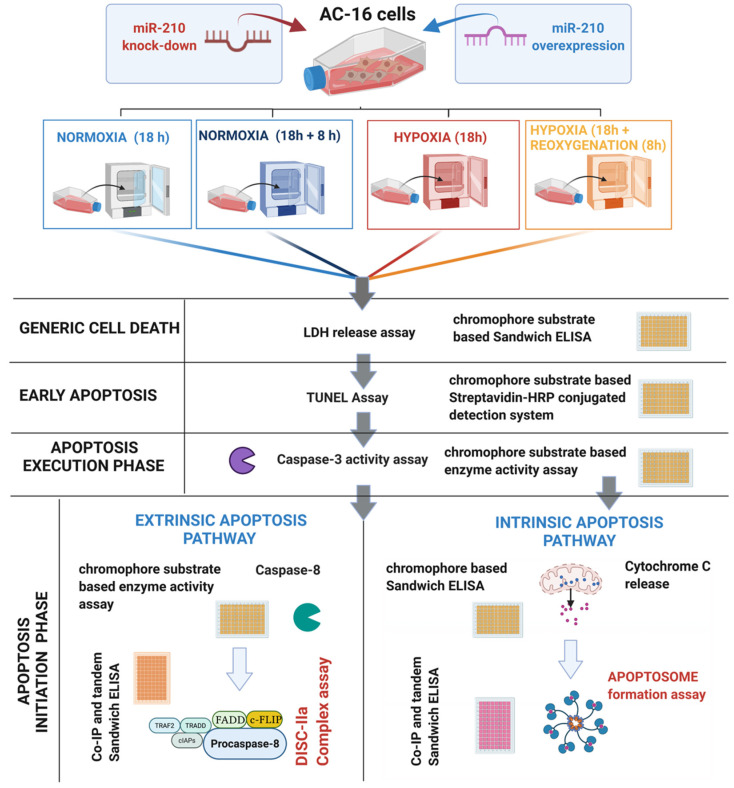
Simplified outline of the methodological workflow.

**Figure 2 biomedicines-10-00042-f002:**
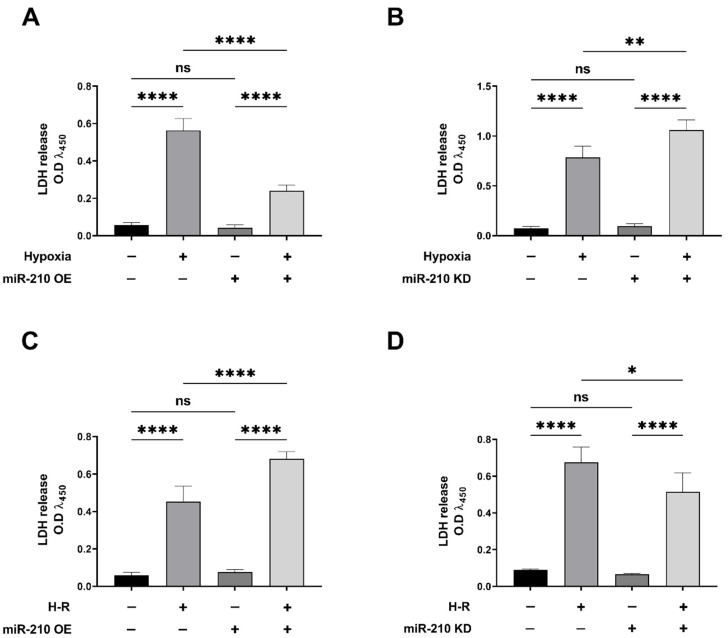
miR-210 attenuated the hypoxia (18 h)-induced cell death but exacerbated the hypoxia-reoxygenation (H-R)-induced cell death. (**A**,**B**) Quantitative ELISA determining the LDH release in the conditioned media as a surrogate marker of cell death unequivocally demonstrated that overexpression of miR-210 significantly mitigated the hypoxia-induced increase in LDH release (**A**) while the ectopic expression of the miR-210-3p decoy/inhibitor significantly augmented the hypoxia-induced increase in LDH release (**B**). (**C**,**D**) Quantitative ELISA unequivocally demonstrated that overexpression of miR-210 significantly enhanced the H-R-induced increase in LDH release (**C**) while the ectopic expression of the miR-210-3p decoy/inhibitor significantly mitigated the H-R-induced increase in LDH release (**D**). miR-210 expression levels in the corresponding respective cell lysates were determined by the miR-210 hybridization immunoassay (as described in the Section 2.3) and are reported in Figure 3, since the respective cell lysates were subjected to the caspase-3 activity assay. Data from the LDH ELISA are expressed as experimental blank-corrected absorbances (O.D) measured at λ_450_ (450 nm). Data are expressed as mean ± S.D from three technical replicates for each of the four biological replicates belonging to each experimental group (*n* = 4). * *p* ≤ 0.05; ** *p* ≤ 0.01; **** *p* ≤ 0.0001; ns: not significant (*p* > 0.05); OE: miR-210 overexpression; KD: miR-210-3p decoy/inhibitor; H-R: hypoxia-reoxygenation O.D: optical density; S.D: standard deviation.

**Figure 3 biomedicines-10-00042-f003:**
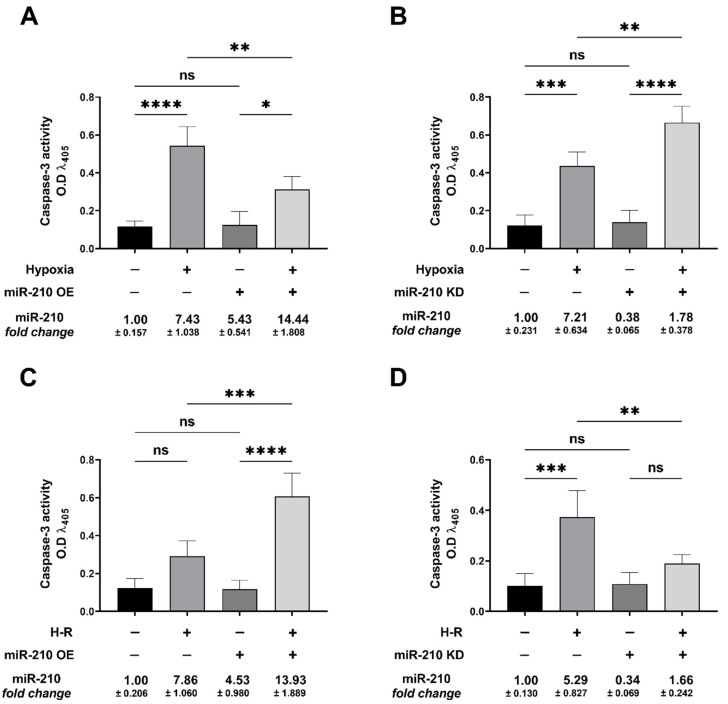
miR-210 mitigated the hypoxia (18 h)-induced increase in *caspase-3* activity while augmenting the hypoxia-reoxygenation (H-R)-induced increase in *caspase-3* activity. (**A**,**B**) Quantitative *caspase-3* activity assays unequivocally demonstrated that overexpression of miR-210 significantly mitigated the hypoxia-induced increase in *caspase-3* activity (**A**), whereas the ectopic expression of the miR-210-3p decoy/inhibitor significantly enhanced the hypoxia-induced increase in *caspase-3* activity (**B**). (**C**,**D**) Quantitative *caspase-3* activity assays demonstrated that overexpression of miR-210 significantly enhanced the H-R-induced increase in *caspase-3* activity (**C**), whereas the ectopic expression of the miR-210-3p decoy/inhibitor significantly mitigated the H-R-induced increase in *caspase-3* activity (**D**). miR-210 expression levels in all experimental groups were determined by the miR-210 hybridization immunoassay, as described in the Section 2.3. Data from the *caspase-3* activity assay are expressed as experimental blank-corrected absorbances (O.D) measured at λ_405_ (405 nm). Data from the *caspase-3* activity assay are expressed as mean ± S.D from three technical replicates for each of the four biological replicates belonging to each experimental group (*n* = 4). miR-210 expression levels are depicted as fold-change ± S.D. * *p* ≤ 0.05; ** *p* ≤ 0.01; *** *p* ≤ 0.001; **** *p* ≤ 0.0001; ns: not significant (*p* > 0.05) OE: miR-210 overexpression; KD: miR-210-3p decoy/inhibitor; H-R: hypoxia-reoxygenation O.D: optical density; S.D: standard deviation.

**Figure 4 biomedicines-10-00042-f004:**
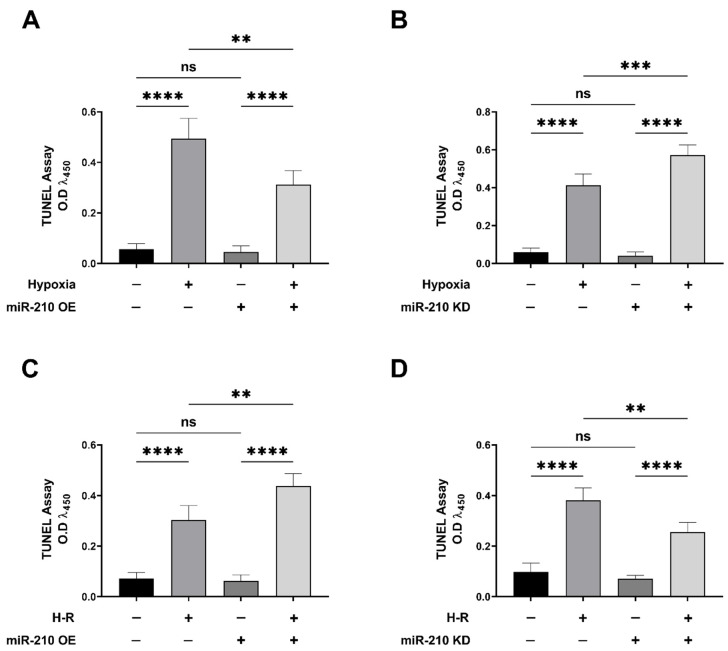
miR-210 attenuated the hypoxia (18 h)-induced *apoptotic DNA fragmentation* while enhancing the hypoxia-reoxygenation (H-R)-induced *apoptotic DNA fragmentation*. (**A**,**B**) Quantitative TUNEL assays determining apoptotic DNA fragmentation unequivocally showed that overexpression of miR-210 significantly mitigated the hypoxia-induced increase in apoptotic DNA fragmentation (**A**), whereas the ectopic expression of the miR-210-3p decoy/inhibitor significantly enhanced the hypoxia-induced increase in apoptotic DNA fragmentation (**B**). (**C**,**D**) Quantitative TUNEL assays determining apoptotic DNA fragmentation unequivocally showed that overexpression of miR-210 significantly augmented the H-R-induced increase in apoptotic DNA fragmentation (**C**), whereas the ectopic expression of the miR-210-3p decoy/inhibitor significantly reduced the H-R-induced increase in apoptotic DNA fragmentation (**D**). Data are expressed as experimental blank-corrected absorbances (O.D) measured at λ_450_ (450 nm). Data are expressed as mean ± S.D from three technical replicates for each of the four biological replicates belonging to each experimental group (*n* = 4); ** *p* ≤ 0.01; *** *p* ≤ 0.001; **** *p* ≤ 0.0001; ns: not significant (*p* > 0.05); OE: miR-210 overexpression; KD: miR-210-3p *decoy/inhibitor*; H-R: hypoxia-reoxygenation; O.D: optical density; S.D: standard deviation.

**Figure 5 biomedicines-10-00042-f005:**
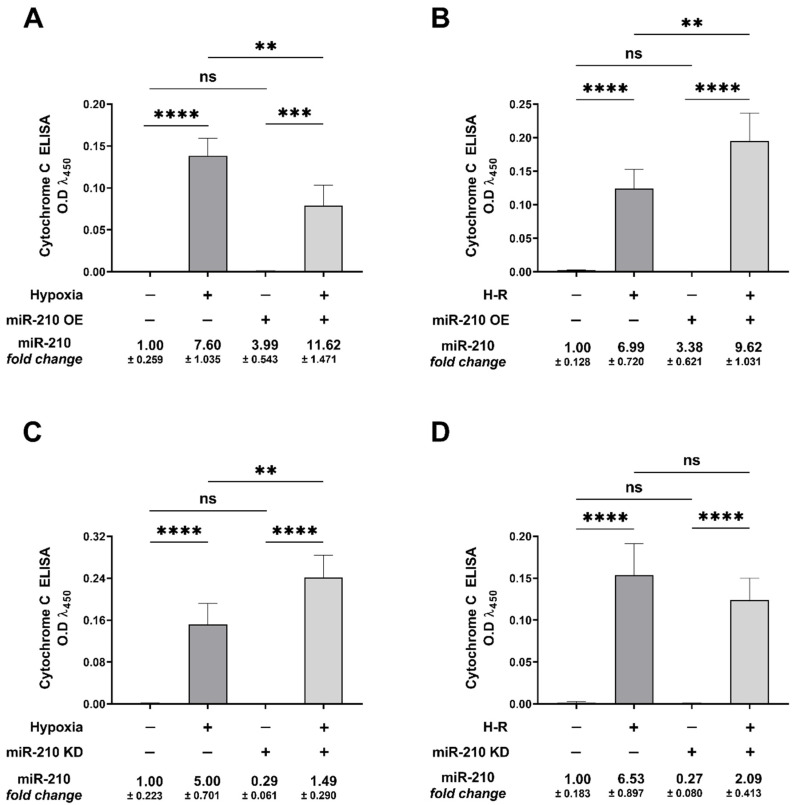
miR-210 attenuated the hypoxia (18 h)-induced increase in *cytochrome c* release, while enhancing the hypoxia-reoxygenation (H-R)-induced increase in *cytochrome c* release. (**A**–**D**) Quantitative ELISA determining the *cytochrome c* abundance in the cytosolic fractions showed that the overexpression of miR-210 significantly reduced the hypoxia-induced increase in *cytochrome c* translocation from the mitochondria into the cytosol (**A**,**C**), whereas the ectopic expression of the miR-210-3p decoy/inhibitor significantly enhanced the hypoxia-induced increase in *cytochrome c* translocation from the mitochondria into the cytosol (**B**,**D**). miR-210 expression levels in all experimental groups were determined by the miR-210 hybridization immunoassay, as described in the Section 2.3. Data from the cytochrome c ELISA is expressed as experimental blank-corrected absorbances (O.D) measured at λ_450_ (450 nm). Data from the *cytochrome c* ELISA are expressed as mean ± S.D from three technical replicates for each of the four biological replicates belonging to each experimental group (*n* = 4). miR-210 expression levels are depicted as fold-change ± S.D; ** *p* ≤ 0.01; *** *p* ≤ 0.001; **** *p* ≤ 0.0001; ns: not significant (*p* > 0.05); OE: miR-210 overexpression; KD: miR-210-3p decoy/inhibitor; H-R: hypoxia-reoxygenation; O.D: optical density; S.D: standard deviation.

**Figure 6 biomedicines-10-00042-f006:**
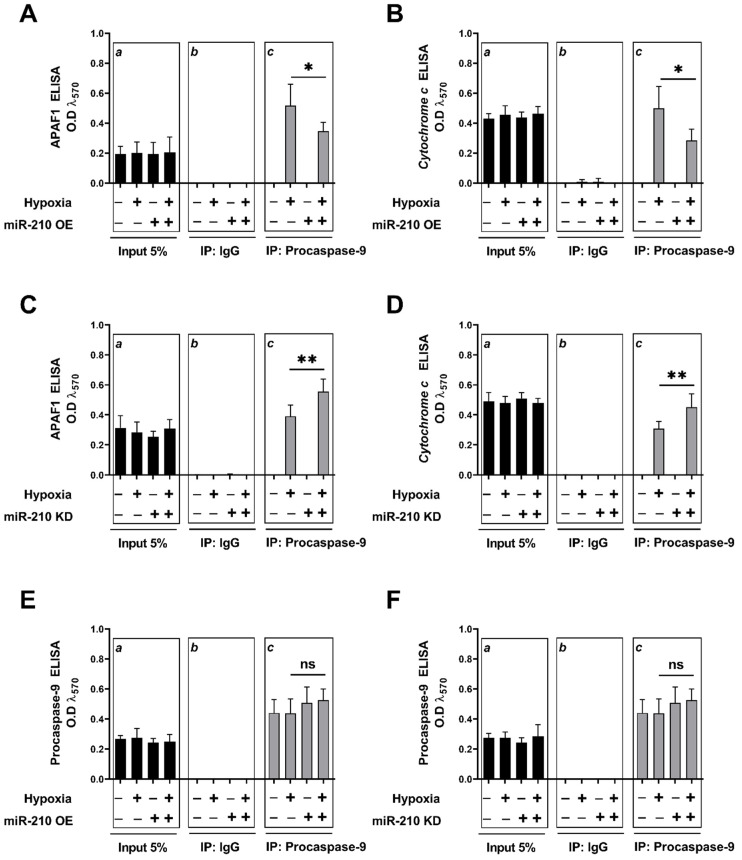
miR-210 reduced the hypoxia (18 h)-induced *apoptosome* formation. (**A**–**F**) Co-immunoprecipitation (Co-IP) coupled with tandem ELISA determining the quantitative abundance of APAF1 and *cytochrome c* in the *procaspase-9* immunoprecipitates were performed as a surrogate measure of the abundance of the *apoptosome* formation. (**A**–**D**) ELISA immunoassays demonstrated that overexpression of endogenous miR-210 significantly attenuated the hypoxia-induced *APAF1* (**A** inset c) and *cytochrome c* (**B** inset c) abundance in the *procaspase-9* immunoprecipitates, whereas the ectopic expression of the miR-210-3p decoy/inhibitor significantly enhanced the hypoxia-induced APAF1 (**C** inset c) and *cytochrome c* (**D** inset c) abundance in *procaspase-9* immunoprecipitates. (**E**,**F**) ELISA determining the quantitative abundance of *procaspase-9* (**E** inset c, **F** inset c) unequivocally showed equitable abundance of *procaspase-9* in the *procaspase-9* immunoprecipitates. ((**A**) inset b—(**F**) inset b) ELISA determining the quantitative abundance of *APAF1* (inset b in (**A**,**C**)), *cytochrome c* (inset b in (**B**,**D**)), and *procaspase-9* (inset b in (**E**,**F**)) in the rabbit *IgG* immunoprecipitates demonstrated the specificity of the *procaspase-9* Co-IP assays. ((**A**) inset a—(**F**) inset a) ELISA determining the quantitative abundance of *APAF1* (inset a in (**A**,**C**)), *cytochrome c* (inset a in (**B**,**D**)), and *procaspase-9* (inset a in (**E**,**F**)) in the native lysates serving as input (5%). miR-210 expression levels in the respective cell lysate-*inputs* were determined by the miR-210 hybridization immunoassay (as described in the Section 2.3) and are reported in Appendix A. Data are expressed as experimental blank-corrected absorbances (O.D) measured at λ_570_ (570 nm). Data are expressed as mean ± S.D from three technical replicates for each of the four biological replicates belonging to each experimental group (*n* = 4); * *p* ≤ 0.05; ** *p* ≤ 0.01; ns: not significant (*p* > 0.05); OE: miR-210 overexpression; KD: miR-210-3p decoy/inhibitor; H-R: hypoxia-reoxygenation; O.D: optical density; S.D: standard deviation.

**Figure 7 biomedicines-10-00042-f007:**
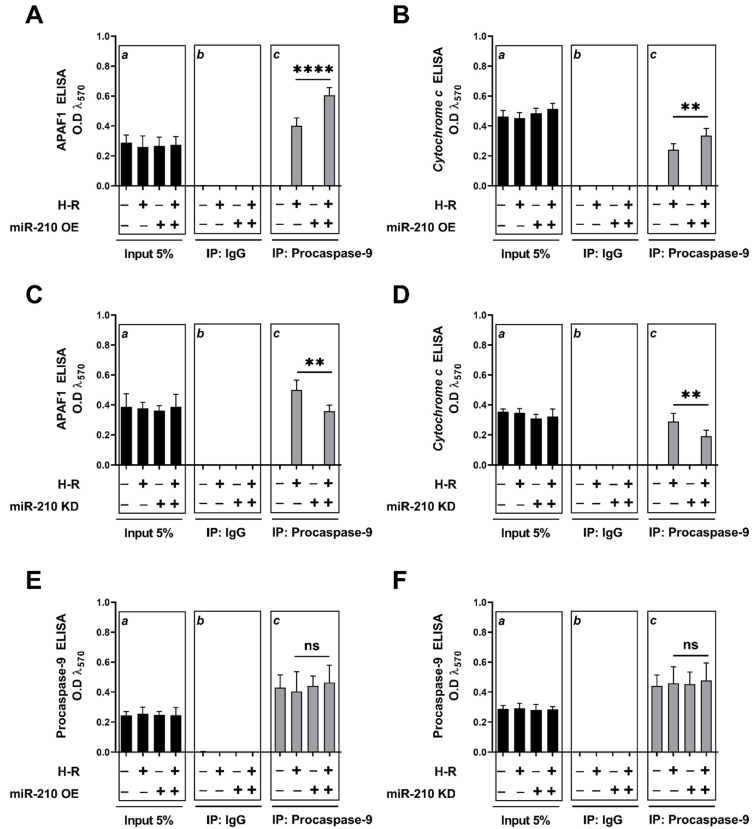
miR-210 exacerbated the hypoxia-reoxygenation (H-R)-induced *apoptosome* formation. (**A**–**D**) ELISA immunoassays demonstrated that overexpression of miR-210 significantly augmented the H-R-induced *APAF1* ((**A**) inset c) and *cytochrome c* ((**B**) inset c) abundance in the *procaspase-9* immunoprecipitates, whereas the ectopic expression of the miR-210-3p decoy/inhibitor significantly attenuated the H-R-induced *APAF1* ((**C**) inset c) and cytochrome c ((**D**) inset c) abundance in *procaspase-9* immunoprecipitates. (**E**,**F**) ELISA determining the quantitative abundance of *procaspase-9* ((**E**) inset c, (**F**) inset c) unequivocally showed equitable abundance of procaspase-9 in the procaspase-9 immunoprecipitates. ((**A**) inset b—(**F**) inset b) ELISA determining the quantitative abundance of *APAF1* (inset b in (**A**,**C**)), *cytochrome c* (inset b in (**B**,**D**)), and *procaspase-9* (inset b in (**E**,**F**)) in the rabbit *IgG* immunoprecipitates demonstrated the specificity of the *procaspase-9* Co-IP assays. ((**A**) inset a—(**F**) inset a) ELISA determining the quantitative abundance of *APAF1* (inset a in (**A**,**C**)), cytochrome c (inset a in (**B**,**D**)), and *procaspase-9* (inset a in (**E**,**F**)) in the native lysates serving as input (5%). miR-210 expression levels in the respective cell lysate-inputs were determined by the miR-210 hybridization immunoassay (as described in the Section 2.3) and are reported in Appendix A. Data are expressed as experimental blank-corrected absorbances (O.D) measured at λ_570_ (570 nm). Data are expressed as mean ± S.D from three technical replicates for each of the four biological replicates belonging to each experimental group (*n* = 4).** *p* ≤ 0.01; **** *p* ≤ 0.0001; ns: not significant (*p* > 0.05); OE: miR-210 overexpression; KD: miR-210-3p decoy/inhibitor; H-R: hypoxia-reoxygenation; O.D: optical density; S.D: standard deviation.

**Figure 8 biomedicines-10-00042-f008:**
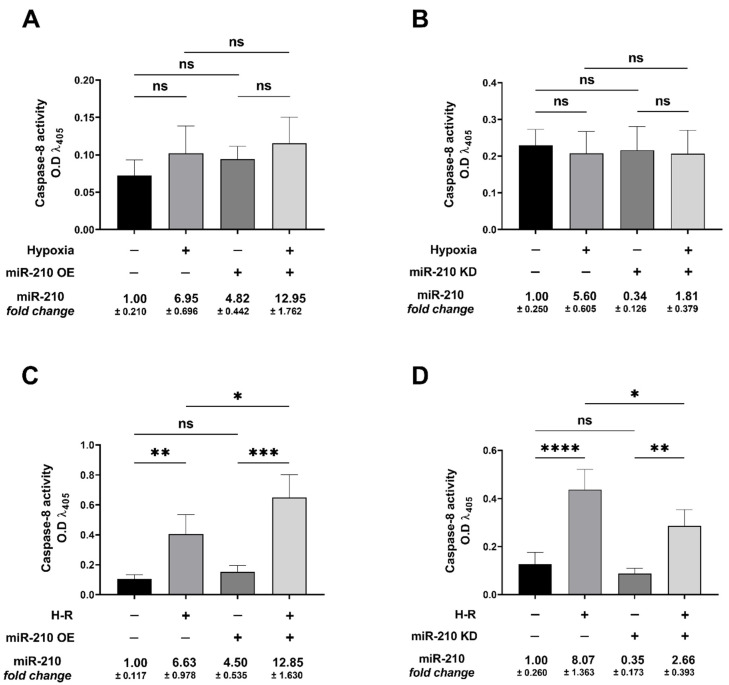
miR-210 increased the hypoxia-reoxygenation (H-R)-induced caspase-8 activation. (**A**,**B**) Quantitative caspase-8 activity assays unequivocally demonstrated that neither hypoxia (18 h) nor the modulation of miR-210 expression during hypoxia (18 h) evoked any changes in caspase-8 activity. (**C**,**D**) Quantitative caspase-8 activity assays demonstrated that overexpression of miR-210 significantly enhanced the H-R-induced increase in caspase-8 activity (**C**), whereas the ectopic expression of the miR-210-3p *decoy/inhibitor* significantly mitigated the H-R-induced increase in caspase-8 activity (**D**). miR-210 expression level in all experimental groups was determined by the miR-210 hybridization immunoassay, as described in the Materials and Methods section. Data from the caspase-8 activity assay are expressed as experimental *blank-corrected* absorbances (O.D) measured at λ_405_ (405 nm). Data from the caspase-8 activity assay are expressed as *mean ± S.D* from three technical replicates for each of the four biological replicates belonging to each experimental group (*n* = 4). miR-210 expression levels are depicted as *fold-change ± S.D;* * *p* ≤ 0.05; ** *p* ≤ 0.01; *** *p* ≤ 0.001; **** *p* ≤ 0.0001; ns: not significant (*p* > 0.05); OE: miR-210 overexpression; KD: miR-210-3p *decoy/inhibitor*; H-R: hypoxia-reoxygenation; O.D: optical density; S.D: standard deviation.

**Figure 9 biomedicines-10-00042-f009:**
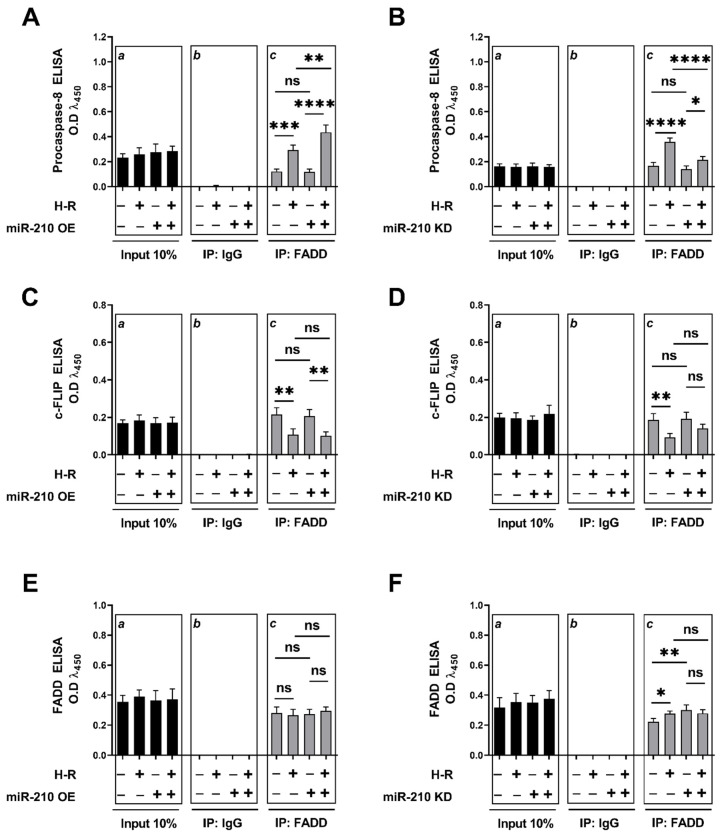
miR-210 increased the hypoxia-reoxygenation (H-R)-induced *DISC-IIa complex* formation, thereby activating the *extrinsic apoptosis* pathway. (**A**–**F**) Co-IP coupled with *tandem* ELISA determining the quantitative abundance of procaspase-8 and c-FLIP in the FADD immunoprecipitates from *RIPK1-*precleared lysates were performed as a surrogate measure of the abundance of the *DISC-IIa complex* formation and the subsequent activation of the *extrinsic apoptosis* pathway. (**A**,**C**) ELISA immunoassays unequivocally show that overexpression of miR-210 significantly increased the H-R-induced procaspase-8 abundance ((**A**) inset c), while having no effect on c-FLIP abundance ((**C**) inset c), in the FADD immunoprecipitates from *RIPK1-*precleared lysates. (**B**,**D**) ELISA immunoassays unequivocally demonstrated that the ectopic expression of the miR-210-3p *decoy/inhibitor* significantly reduced the H-R-induced procaspase-8 abundance (C inset c), while eliciting no effect on c-FLIP abundance ((**D**) inset c), in the in the FADD immunoprecipitates from *RIPK1*-precleared lysates. (**E**,**F**) ELISA determining the quantitative abundance of *FADD* ((E) inset c, (**F**) inset c) in the *FADD* immunoprecipitates from *RIPK1-precleared lysates*. ((**A**) inset b—(**F**) inset b) ELISA showing the absence of *procaspase-8* (inset b in (**A**,**B**), *c-FLIP* (inset b in (**C**,**D**), and *FADD* (inset b in (**E**,**F**)) in the mouse *IgG* immunoprecipitates from *RIPK1-precleared lysates*, thereby demonstrating the specificity of the *FADD* Co-IP assays. ((**A**) inset a—(**F**) inset a) ELISA determining the quantitative abundance of procaspase-8 (inset a in (**A**,**B**), c-FLIP (inset a in (**B**,**D**)), and FADD (inset a in (E,F) in the *RIPK1* precleared lysates serving as input (10%). The validity and integrity of the *RIPK1-*precleared lysates in all experimental groups were determined by ELISA (as described in Section 2.11) and are reported in Appendix A. miR-210 expression levels in the respective cell lysate inputs were determined by the miR-210 hybridization immunoassay (as described in the Section 2.3) and are reported in Appendix A. Data are expressed as experimental blank-corrected absorbances (O.D) measured at λ_450_ (450 nm). Data are expressed as mean ± S.D from three technical replicates for each of the four biological replicates belonging to each experimental group (*n* = 4).* *p* ≤ 0.05; ** *p* ≤ 0.01; *** *p* ≤ 0.001; **** *p* ≤ 0.0001; ns: not significant (*p* > 0.05); OE: miR-210 overexpression; KD: miR-210-3p decoy/inhibitor; H-R: hypoxia-reoxygenation O.D: optical density; S.D: standard deviation.

**Figure 10 biomedicines-10-00042-f010:**
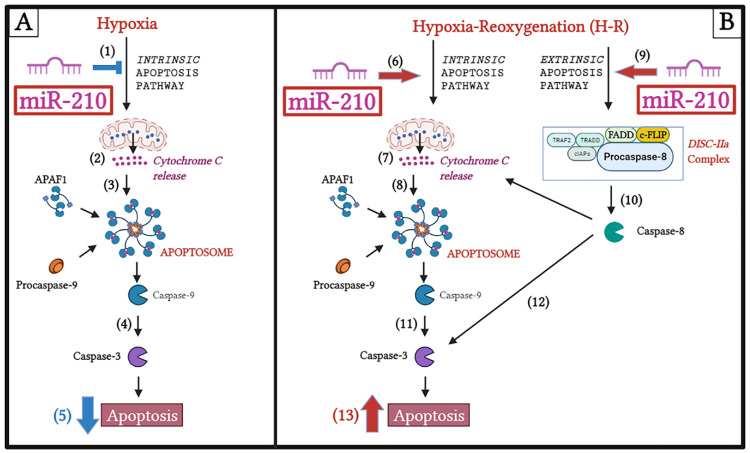
An illustrative model of the biphasic *Janus* role of microRNA-210 (miR-210) in modulating apoptotic cell death during cellular hypoxia and reoxygenation phases. (**A**) *miR-210 inhibited the hypoxia-induced intrinsic apoptosis pathway* (1). miR-210 inhibited the hypoxia-induced *cytochrome c* translocation from the mitochondria to the cytosol (2), resulting in a decrease of the abundance of the *apoptosome complex* (3), the protein complex that serves as the rate-limiting factor in the caspase cascade activation. The decrease in the abundance of the *apoptosome complex* translates into a decrease flux in *caspase-9* activation (not determined in this study) that results in the attenuation of the downstream *effector*, *caspase-3* activity (4). The miR-210-elicited mitigation in hypoxia-induced *caspase-3* activity culminated in the attenuation of hypoxia-induced apoptotic cell death (5). (**B**) *miR-210 augmented the hypoxia-reoxygenation (H-R)-induced flux through the intrinsic apoptosis pathway (6), as well as the extrinsic apoptosis pathway* (9). miR-210 exacerbated the H-R-induced *cytochrome c* release into the cytosol (7), resulting in an increase in the *apoptosome complex* formation (8). miR-210 also fostered the H-R-induced *DISC-IIa complex* formation (9), the protein complex that characterizes the rate-limiting step in the flux through *extrinsic apoptosis pathway*. The increase in the abundance of the *DISC-IIa complex* translated into an increase flux in *caspase-8* activation (10). The miR-210 exacerbation of the H-R-induced *apoptosome complex* formation (8) and *caspase-8* activation (10) resulted in a significant increase in *caspase-3* activity (11,12). The miR-210-elicited exacerbation of H-R-induced *caspase-3* activity culminated in the augmentation of H-R-induced apoptotic cell death (13). This illustration was created in BioRender.com and adapted from “Apoptosis Extrinsic and Intrinsic Pathways” by BioRender.com (2021). Retrieved from https://app.biorender.com/illustrations/61826429218a0b00a6f7b838 on 3 November 2021.

**Table 1 biomedicines-10-00042-t001:** Hypoxia and hypoxia + reoxygenation experimental paradigm.

	Control Empty Vector (pEZX-MR04-Scrambled)	miR-210 Overexpression (OE) Vector (pEZX-MR04-miR-210)	Control Empty Vector (pEZX-AM01- Scrambled)	miR-210 Decoy/Inhibition (KD) Vector (pEZX-AM01-miR-210)
Normoxia, 18 h	*n* = 4	*n* = 4	*n* = 4	*n* = 4
Hypoxia, 18 h	*n* = 4	*n* = 4	*n* = 4	*n* = 4
Normoxia (18 + 8 h)	*n* = 4	*n* = 4	*n* = 4	*n* = 4
Hypoxia (18 h) + Reoxygenation (8 h)	*n* = 4	*n* = 4	*n* = 4	*n* = 4

*n* = 4: four biological replicates.

**Table 2 biomedicines-10-00042-t002:** List of antibodies used in the study.

Antibody	Application	Amount	Host	Manufacturer	Catalogue	Resource Identifier ID (RRID)
β-Actin	WB 1:5000	1 µg	Mouse	Santa Cruz Biotechnology	sc-47778	AB_2714189
APAF1	IP	5 µg	Rabbit	Novus Biologicals	NBP1-77000	AB_11008194
APAF1	ELISA *capture*	50 ng/well	Mouse	Santa Cruz Biotechnology	sc-65891	AB_1119006
APAF1	ELISA *detection*	50 ng/well	Rabbit	Novus Biologicals	NBP1-77000	AB_11008194
c-FLIP	ELISA *capture*	50 ng/well	Mouse	Santa Cruz Biotechnology	sc-5276	AB_627764
c-FLIP	ELISA *detection*	50 ng/well	Rabbit	Novus Biologicals	NBP1-77016	AB_11024867
Cytochrome c	WB 1:1000	5 µg	Mouse	Thermo Fisher	BMS1037	AB_10598651
Cytochrome c	ELISA *capture*	20 ng/well	Mouse	Thermo Fisher	BMS1037	AB_10598651
Cytochrome c	ELISA *detection*	20 ng/well	Rabbit	Cell Signaling Technology	4280	AB_10695410
Procaspase-8/Caspase-8	ELISA *capture*	40 ng/well	Mouse	Cell Signaling Technology	9746	AB_2275120
Procaspase-8/Caspase-8	ELISA *detection*	40 ng/well	Rabbit	Novus Biologicals	NBP1-76610	AB_11034997
Procaspase-8/Caspase-8	WB 1:1000	5 µg	Mouse	Cell Signaling Technology	9746	AB_2275120
Procaspase-9/Caspase-9	IP	5 µg	Rabbit	Cell Signaling Technology	9502	AB_2068621
Procaspase-9/Caspase-9	ELISA *capture*	40 ng/well	Mouse	Cell Signaling Technology	9508	AB_2068620
Procaspase-9/Caspase-9	ELISA *detection*	40 ng/well	Rabbit	Novus Biologicals	NBP1-76961	AB_11034844
FADD	IP	5 µg	Mouse	Sigma Aldrich/Merck Life Science	F8053	AB_476989
FADD	ELISA *capture*	30 ng/well	Rabbit	Cell Signaling Technology	2782	AB_2100484
FADD	ELISA *detection*	30 ng/well	Mouse	BioVision/VWR	3039-100/ 10005-490	AB_2100612
Goat Anti-Mouse IgG (H + L)-HRP Conjugate	1:5000	1 µg	Goat	Bio-Rad	1706516	AB_11125547
Goat Anti-Mouse IgG-AP Conjugate	1:5000	N/A ^€^	Goat	Bio-Rad	1706520	AB_11125348
Goat Anti-Rabbit IgG (H + L)-HRP Conjugate	1:5000	1 µg	Goat	Bio-Rad	1706515	AB_11125142
Goat Anti-Rabbit IgG-AP Conjugate	1:20,000	N/A ^€^	Goat	Sigma Aldrich/Merck Life Science	A3687	AB_258103
LDH	ELISA *capture*	30 ng/well	Mouse	Santa Cruz Biotechnology	sc-133123	AB_2134964
LDH-A	ELISA *detection*	30 ng/well	Rabbit	Novus Biologicals	NBP1-48336	AB_10011099
LDH-B	ELISA *detection*	30 ng/well	Rabbit	Novus Biologicals	NBP2-38131	N/A
Mouse IgG	IP	5 µg	Mouse	Santa Cruz Biotechnology	sc-2025	AB_737182
Rabbit IgG	IP	5-10 µg	Rabbit	Santa Cruz Biotechnology	sc-3888	AB_737196
RIPK1	IP	10 µg	Rabbit	Cell Signaling Technology	3493	AB_2305314
RIPK1	ELISA *capture*	5 µg	Mouse	Santa Cruz Biotechnology	sc-133102	AB_1568814
RIPK1	ELISA *detection*	50 ng/well	Rabbit	Sigma Aldrich/Merck Life Science	SAB3500420	AB_10643987
TOM20	1:5000	1 µg	Mouse	Thermo Fisher Scientific	MA5-34964	AB_2848869

WB: Western blot; IP: immunoprecipitation; N/A: not available; ^€^: amount of secondary antibody cannot be determined as the commercial vendor does not provide the antibody concentration.

## Data Availability

All data included in the manuscript. Supporting information could be requested from the corresponding author.

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
