# Peer review of "miR-210 Regulates Apoptotic Cell Death during Cellular Hypoxia and Reoxygenation in a Diametrically Opposite Manner"

_biomedicines, 2021, doi:10.3390/biomedicines10010042_

Round 1

Reviewer 1 Report

The manuscript by Gurdeep Marwarha et al.

Title: miR-210 regulates apoptotic cell death during cellular hypoxia and reoxygenation in a diametrically opposite manner

Main result: Results suggested a bimodal effect of miR-210, a master hypoxamiR, on apoptotic cell death. The study demonstrated that miR-210, attenuates the hypoxia-driven intrinsic apoptosis pathway, while significantly augmenting the reoxygenation-induced caspase-8 – mediated extrinsic apoptosis pathway.

The study is of importance for scientific community. The manuscript is well written. I have few comments.

Limitations of the study should be more clearly described.

Authors should consider to add a short discussion regarding miR-210 target genes.

Future directions should be added.

A graphical figure summarizing methods and main results would attract citations and improve the study.

Author Response

Limitations of the study should be more clearly described.

We thank the reviewer for asking for this. We have now added a section with limitation at the end of the discussion

Authors should consider to add a short discussion regarding miR-210 target genes.

We have now added a section on this in the discussion.

Future directions should be added.

This is now added to the discussion.

A graphical figure summarizing methods and main results would attract citations and improve the study.

We thank the reviewer for requesting this figure. We do absolutely agree that this will improve the study. We have added this figure of methodological workflow at the end of the methods chapter. 

Reviewer 2 Report

The manuscript on the regulation of apoptotic cell death during hypoxia and pre-oxygenation by miR20 includes very interesting topics.

It covers the most important issues in this area.

It is constructed very well.

It is noteworthy that miR20 attenuates hypoxia induced intrinsic apoptosis pathway while reoxygenation augmenting extrinsic apoptosis pathway.

The results are backed up by many experiences with well-chosen groups.

I have no objections to the methodology.

Well-designed experiences are a strong point of the manuscript.

I have only few minor comments:

Maybe it would be good to change the words “Hypoxia-Reoxygenation” on “Hypoxia + Reoxygenation” in Table 1 to emphasize the duration of these stages 18hours + 8hours.

It may also be better to use “hypoxia-reoxygenation” in the legend for the figures instead of the abbreviation “H-R”.

Author Response

I have only few minor comments:

Maybe it would be good to change the words “Hypoxia-Reoxygenation” on “Hypoxia + Reoxygenation” in Table 1 to emphasize the duration of these stages 18hours + 8hours.

We agree with the reviewer and have now changes this in the revised manuscript

It may also be better to use “hypoxia-reoxygenation” in the legend for the figures instead of the abbreviation “H-R”.

This is now changed according to the reviewers suggestions.